# Multiplexed CRISPR/CAS9-mediated engineering of pre-clinical mouse models bearing native human B cell receptors

Xuesong Wang[1,†], Rashmi Ray[1,†], Sven Kratochvil[1], Eleonora Melzi[1] [ID], Ying-Cing Lin[1], Sophie Giguere[1], Liling Xu[1], John Warner[1], Diane Cheon[1], Alessia Liguori[2,3,4], Bettina Groschel[2,3,4], Nicole Phelps[2,3,4], Yumiko Adachi[2,3,4], Ryan Tingle[2,3,4], Lin Wu[5], Shane Crotty[4,6,7], Kathrin H Kirsch[1] [ID], Usha Nair[1] [ID], William R Schief[1,2,3,4,*] [ID] & Facundo D Batista[1,8,9,**] [ID]

## Abstract

B-cell receptor (BCR) knock-in (KI) mouse models play an important role in vaccine development and fundamental immunological studies. However, the time required to generate them poses a bottleneck. Here we report a one-step CRISPR/Cas9 KI methodology to combine the insertion of human germline immunoglobulin heavy and light chains at their endogenous loci in mice. We validate this technology with the rapid generation of three BCR KI lines expressing native human precursors, instead of computationally inferred germline sequences, to HIV broadly neutralizing antibodies. We demonstrate that B cells from these mice are fully functional: upon transfer to congenic, wild type mice at controlled frequencies, such B cells can be primed by eOD-GT8 60mer, a germline-targeting immunogen currently in clinical trials, recruited to germinal centers, secrete class-switched antibodies, undergo somatic hypermutation, and differentiate into memory B cells. KI mice expressing functional human BCRs promise to accelerate the development of vaccines for HIV and other infectious diseases.

**Keywords** antibody; B cell receptor; CRISPR; Cas9; HIV vaccine; knock-in
**Subject Category** Immunology
**The EMBO Journal (2021) 40: e105926**

## Introduction

Humoral immunity defends against infections and plays a key role in immunization via the secretion of protective antibodies and production of long-lived memory B cells. The elicitation of protective antibodies through vaccination is dependent on a broad range of immunological principles, including: the activation of naïve B cells, their entry into germinal centers (GCs), affinity maturation of the B-cell receptors (BCRs) by iterative somatic hypermutation (SHM) dependent on T follicular cell help ($T_{FH}$) within GCs, and ultimately the differentiation of GC B cells into long-lived memory B cells (MBCs) that can be recalled upon antigen re-exposure, and plasma cells (PCs) that secrete antibodies (Alt et al, 1992; Rajewsky, 1996; McHeyzer-Williams & McHeyzer-Williams, 2005; Vinuesa et al, 2005; Neuberger, 2008; Cyster, 2010; Crotty, 2011; Victora & Nussenzweig, 2012; Crotty, 2014). Although we draw upon these principles in order to successfully generate vaccines, a deeper understanding of the mechanisms that determine antibody specificity, diversification, affinity maturation, and durability is important in developing vaccines against antigenically diverse viruses such as HIV-1 and influenza (Burton et al, 2012; Haynes & Mascola, 2017).

A key aim for vaccines against antigenically diverse viruses is to induce broadly neutralizing antibodies (bnAbs) capable of neutralizing diverse isolates. In the case of HIV-1, bnAbs develop in a small fraction of infected individuals, and passive transfer of bnAbs can prevent infection in animal models (Gauduin et al, 1997; Mascola et al, 2000; Parren et al, 2001; Moldt et al, 2012; Pietzsch et al, 2012; Shingai et al, 2014). Hence, it is widely accepted that vaccine

1  The Ragon Institute of Massachusetts General Hospital, Massachusetts Institute of Technology and Harvard University, Cambridge, MA, USA
2  Department of Immunology and Microbiology, The Scripps Research Institute, La Jolla, CA, USA
3  IAVI Neutralizing Antibody Center, The Scripps Research Institute, La Jolla, CA, USA
4  Consortium for HIV/AIDS Vaccine Development (CHAVD), The Scripps Research Institute, La Jolla, CA, USA
5  Genome Modification Facility, Department of Molecular and Cellular Biology, Harvard University, Cambridge, MA, USA
6  Center for Infectious Disease and Vaccine Research, La Jolla Institute for Immunology (LJI), La Jolla, CA, USA
7  Department of Medicine, University of California, San Diego, La Jolla, CA, USA
8  Department of Immunology, Harvard Medical School, Boston, MA, USA
9  Department of Microbiology, Harvard Medical School, Boston, MA, USA
   *Corresponding author. Tel: +1 858 784 7725; E-mail: schief@scripps.edu
   **Corresponding author. Tel: +1 857 268 7071; E-mail: fbatista1@mgh.harvard.edu
   †These authors contributed equally to this work

induction of bnAbs with sufficient potency, breadth, and durability has potential to protect humans against HIV-1 (Burton *et al*, 2012; Burton & Hangartner, 2016; Kwong & Mascola, 2018; Haynes *et al*, 2019). However, while bnAbs bind HIV Env with high affinity, their unmutated precursors typically lack measurable affinity for most Env isolates. Germline-targeting vaccine design aims to alleviate this problem by engineering immunogens with affinity for unmutated bnAb precursors, which could initiate bnAb generation in the HIV seronegative human population (Jardine *et al*, 2013; McGuire *et al*, 2013; Dosenovic *et al*, 2015; Jardine *et al*, 2015; Briney *et al*, 2016; Escolano *et al*, 2016; McGuire *et al*, 2016; Sok *et al*, 2016; Steichen *et al*, 2016; Tian *et al*, 2016; Jardine *et al*, 2016a; Jardine *et al*, 2016b; Steichen *et al*, 2019).

Pre-clinical testing of germline-targeting immunogens has often relied on knock-in (KI) mouse models expressing inferred germline (iGL) reverted precursors of known bnAbs (Dosenovic *et al*, 2015; Jardine *et al*, 2015; Briney *et al*, 2016; Escolano *et al*, 2016; McGuire *et al*, 2016; Steichen *et al*, 2016; Tian *et al*, 2016; Andrabi *et al*, 2019; Saunders *et al*, 2019; Steichen *et al*, 2019). While this approach has significantly contributed to the field and yielded several important insights, predicting the germline sequence of a highly mutated bnAb in the absence of longitudinal bnAb lineage sequencing data is challenging, in particular for the non-templated regions of the BCR. Furthermore, this raises the question of whether native, human BCRs will respond to germline-targeting Env immunogens in the same manner as iGLs.

Recent studies in which the germline-targeting immunogen eOD-GT8 60mer was successfully employed as a bait to isolate native human B-cell precursors of the VRC01-class bnAbs family from HIV-1 seronegative volunteers provide the framework for addressing the question (Jardine *et al*, 2016a; Havenar-Daughton *et al*, 2018a; Havenar-Daughton *et al*, 2018b). VRC01-class bnAbs target the CD4-binding site of HIV-1 Env and include some of the broadest and most potent bnAbs that have been identified to date (Wu *et al*, 2010; Scheid *et al*, 2011; Wu *et al*, 2011; Huang *et al*, 2016; Sajadi *et al*, 2018). VRC01-class antibodies rely on a heavy chain (HC) using the IgHV1-2 gene, which structurally mimics CD4, a short 5-amino acid LCDR3 motif, and a flexible LCDR1, which is important to avoid a steric clash with gp120 (Zhou *et al*, 2010; Zhou *et al*, 2015). However, these Abs can accommodate a variety of different CDRH3s and LC V-genes, including $V\kappa3$-20, $V\kappa1$-33, $V\kappa1$-5, and $V_L2$-14 (Havenar-Daughton *et al*, 2018b; Umotoy *et al*, 2019).

Seminal work carried out with earlier versions of the VRC01-class germline-reverted HC mouse models have revealed several important insights regarding the ability of tailored immunogens to drive affinity maturation (Dosenovic *et al*, 2015; Jardine *et al*, 2015; Briney *et al*, 2016). However, these studies did not initially address VRC01-class responses at physiologically relevant B-cell precursor frequencies. More recently, these issues have been overcome by taking advantage of a congenic adoptive transfer model, allowing experiments to be performed at well-defined precursor frequencies and affinities imparting relevance to the pre-clinical mouse models (Abbott *et al*, 2018; Dosenovic *et al*, 2018). However, a hurdle that now remains to be overcome is the time required for KI mouse generation. The mouse models generated thus far have exclusively relied on traditional embryonic stem (ES) cell technology, which suffers from the drawback that multiple crossings are required for the full germline transmission of insertions or deletions.

To alleviate this barrier, we previously developed a one-step CRISPR/Cas9-induced homology-directed repair (HDR) approach. We generated mouse models bearing human, pre-rearranged Ig precursors for two different HIV bnAbs (PGT121$^{gH}$ or BG18$^{gH}$) integrated at the endogenous mouse Ig H locus, in a matter of weeks (Lin *et al*, 2018; Steichen *et al*, 2019). In the PGT121$^{gH}$ KI mouse we observed a lack of PGT121 HC bearing B cells in the peripheral blood, indicating that B cells bearing PGT121$^{gH}$ may be auto- or poly-reactive in mice (Lin *et al*, 2018). In contrast, we found that BG18$^{gH}$ precursors could be stimulated after immunization; they accumulated SHM and gave rise to specific antibody responses (Steichen *et al*, 2019). However, at that point we were only successful in generating Ig H chain KI models, wherein the KI H chains paired with mouse Ig L chains. Previously, BCR KI mice expressing bicistronic Ig H and L chains targeted to the Ig H locus had been generated, but a rapid method to generate KI mice expressing Ig L chains at the Ig κ locus does not exist (Jacobsen *et al*, 2018). Therefore, in order to render the Ig KI model physiologically relevant, we extended this approach to generate Ig L chains at the mouse κ locus.

Here, we report for the first time a protocol for the rapid generation of Ig L chains in the Igκ locus. Moreover, we extended this one-step CRISPR/Cas9 technology to multiplex the insertion of human germline Ig H and L chains at the Ig H and Ig κ native mouse loci, respectively. This protocol relies on the concomitant insertion of two large pieces of donor DNA at desired genomic loci of fertilized zygotes at high frequencies. We demonstrate the power of this technology by generating three novel KI mouse models expressing genuine human VRC01-class precursors isolated from healthy human donors (Havenar-Daughton *et al*, 2018b). We show that B cells expressing these VRC01-class BCRs can be primed *in vivo* by eOD-GT8 60mer, recruited to germinal centers even at rare precursor frequency, secrete class-switched antibodies, and undergo somatic hypermutation. Thus, these KI mice are highly valuable in evaluating immune response elicited by vaccine more authentically. It is a major step forward that not only reduces the time for HIV preclinical validation but is also a promising way to accelerate vaccine design for HIV based on the prime-boost strategy.

## Results

### Generation of KI mice expressing human light chain in one-step using CRISPR/Cas9

We have previously shown that using the CRISPR/Cas9 technology, we can accelerate the generation of KI mice expressing pre-rearranged, human germline-reverted Ig H chains at the native Ig H locus, via donor template-mediated homology-directed repair (HDR) zygote microinjection (Lin *et al*, 2018; Steichen *et al*, 2019). However, since BCR specificity is not conferred by the heavy chain alone, in this work we attempted to investigate if we can extend this methodology to generate KI mice expressing similar Ig L chains at the mouse κ locus. Indeed, the rapid generation of mouse models with Ig H and L chains knocked into their native loci would not only facilitate the generation of proper BCR specificity, but also allow the interrogation of class switch recombination, somatic hypermutation (SHM), and affinity maturation in response to infection and immunization under physiological conditions.

We constructed donor plasmids with 3.9 kb 5′ and 3′ homology arms, C57/BL6 mouse Vκ4-53 promoter and its related leader region (1.5 kb), as well as a pre-assembled L chain (0.4 kb) expressing PGT121-gL, the predicted germline VJ sequence of the PGT121 monoclonal antibody light chain (Fig 1A). This sequence, previously described as PGT121 GL$_{CDR3rev1}$ light chain, has been fully germline-reverted (Steichen *et al*, 2016). In order to introduce our KI DNA target in the correct genomic locus and drive homologous recombination, our strategy relied on using Cas9 nuclease to introduce double-stranded breaks specifically at the Jκ1-Jκ5 region of the native Igκ locus of fertilized oocytes. We initiated double-stranded breaks using two single guide RNAs (sgRNAs) as indicated in

Fig 1A. We used the CRISPR DESIGN database (https://zlab.bio/guide-design-resources) to design-specific guide RNAs to cleave precise regions of the native murine Igκ locus of fertilized oocytes. We tested 12 sgRNAs and selected two sgRNAs, sgRNA11 and sgRNA18, based on their efficiency to cleave a PCR amplicon containing the wild type (WT) genomic DNA target in an *in vitro* assay (Fig 1B). An additional reason for selecting these sgRNAs was that the CRISPR DESIGN database predicted minimal off-target effects on related genes. The selected sgRNAs, Cas9 protein, and template plasmid were injected into fertilized oocytes, which were subsequently implanted into pseudo-pregnant female mice. Thirty founder mice (F0) were born, of which seven carried the PGT121 κ

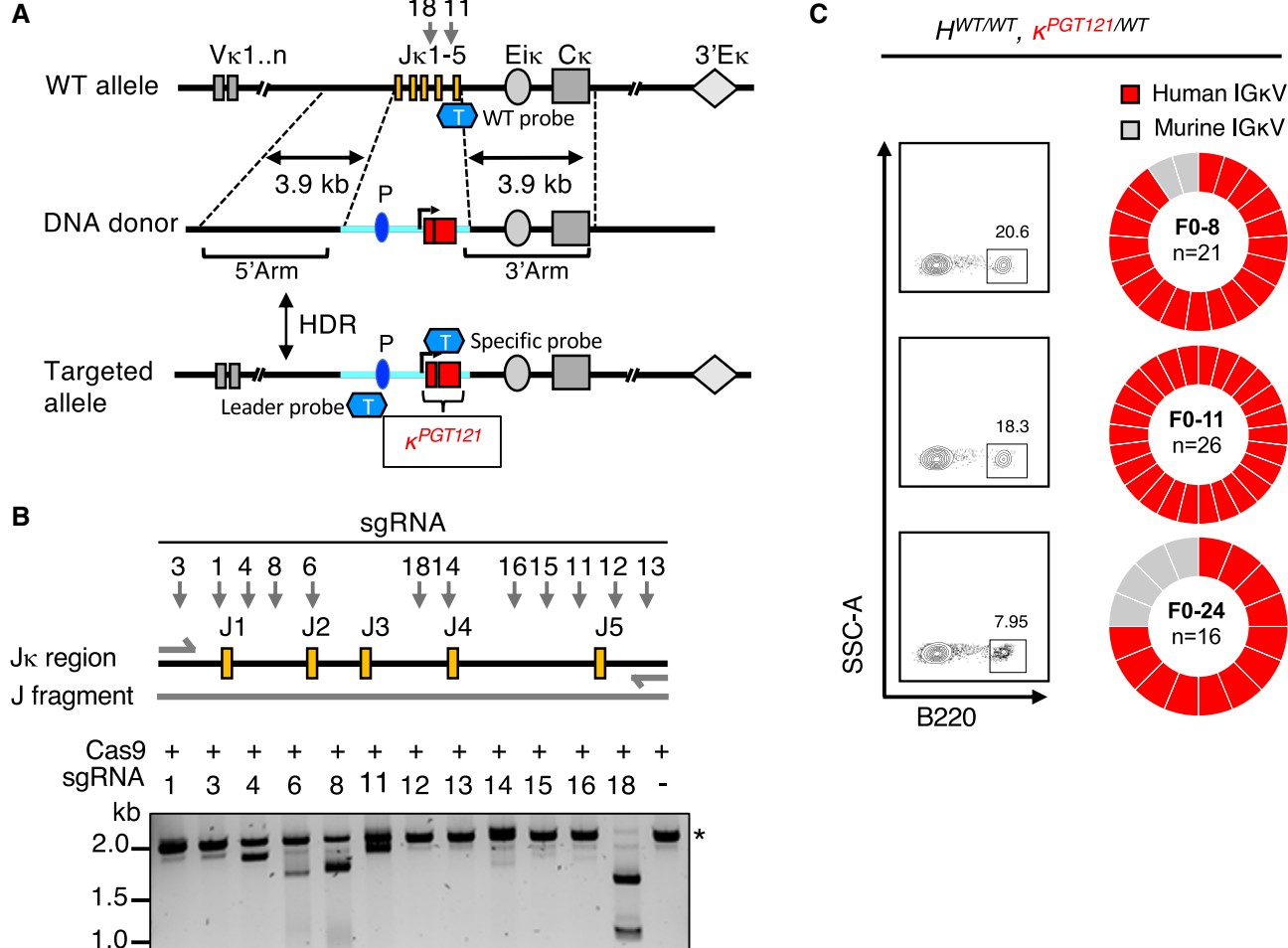

**Figure 1.  Generation of a KI mouse model bearing a pre-rearranged PGT121 κ.**

A   Strategy for the insertion of PGT121 pre-rearranged VJ into the mouse Ig κ locus. Targeting DNA donor with 5′ (3.9 kb) and 3′ (3.9 kb) homology arms to the C57BL/6 WT mouse Ig κ locus, murine promoter, leader, and the human PGT121 light chain VJ sequences are located between two homology arms. Two sgRNAs, 18 and 11, were targeted at J4-J5 region of Ig κ locus. CRISPR/Cas9-mediated HDR leads to the insertion of the promoter and PGT121 sequences into the C57BL/6 mouse genome. V segments, enhancers, and the kappa constant regions are shown in gray and labeled appropriately. Yellow rectangles represent J segments; dark blue oval represents the Vκ4-53 promoter (P); light blue line represents the inserted segment and red rectangles show the rearranged PGT121 VJ. "T" represents TaqMan probe. WT probes were used for the detection of WT allele, Leader probes were used for the detection of the 5′end of the insertion, the specific probes were used for the detection of pre-arranged VJ insertion for PGT121 (probe sequences, see Appendix Table S1).

B   A fragment of genomic DNA (2.2 kb) was amplified by PCR and *in vitro* sgRNA-guided Cas9-mediated cleavage assay was performed with each of the sgRNAs. sgRNA-targeting sites are indicated by arrows, genomic DNA size is indicated by asterisk.

C   B220⁺ single B cells from peripheral blood of three PGT121 LC KI naïve mice were sorted. B220⁺ B-cell populations and their frequencies are shown in FACS plots (left panel). Ig light chains from single-cell sorted B cells were PCR amplified and sequenced. The resulting IGLV libraries were compared to the PGT121 LC reference sequence. The pie charts indicate the frequency of IGLV sequences identical to human PGT121 (red) and mouse IGLV (gray).

chain insertion (23.3% success rate) by genotyping using TaqMan probes (Fig 1A and Appendix Fig S1A, Appendix Tables S1 and S2). Of the seven F0 founders, in six of them the PGT121 κ chain insertion was present in one Ig κ allelic locus, whereas the other Ig κ locus was wild type ($\kappa^{PGT121/WT}$); in the last animal (F0–15), the PGT121 κ chain was inserted in one Ig κ allelic locus and the WT Jκ1-Jκ5 segment was deleted in the other allele (Appendix Table S2). We further crossed all of our positive light chain KI F0 founders with WT mice and followed the frequency of germline transmission of the KI light chain. Although the numbers of F1 progeny were not large, four out of seven F0 mice appeared to transmit the KI PGT121 κ chain in a Mendelian fashion.

Next, to confirm the expression of the PGT121 κ chain within the cell-surface BCR, we isolated and single-cell sorted B220$^+$ peripheral B cells (Live/CD4$^-$ CD8$^-$ Gr1$^-$ F4/80$^-$ B220$^+$) from three F0 $\kappa^{PGT121/WT}$ mice, and determined the frequency of PGT121 κ chain germline sequences via BCR sequencing. In these three $\kappa^{PGT121/WT}$ animals, ~ 90.5% (57 out of 63) of the κ chain sequences were identical to the original PGT121 κ chain germline sequence, whereas ~ 9.5% (6 out of 63) were wild type (Fig 1C). We also obtained single-cell paired H-L chain sequences from B220$^+$ peripheral B cells of the three F1 $\kappa^{PGT121/WT}$ mice. We observed that PGT121 κ chain was paired with mouse Ig HC, suggesting that the PGT121 κ chain was part of a functional BCR. Furthermore, the Ig H V gene family usage also appeared to be more or less evenly distributed, and not skewed toward any one particular Ig H V gene (Appendix Fig S1B). Our results reveal that using the CRISPR/Cas9 technology, we are able to knock in a relatively large DNA fragment (~ 2 kb) containing the pre-rearranged VJ kappa to generate F0 animals in a matter of weeks, and that the knock-in allele is transmitted to the F1 progeny. In this test case, the CRISPR/Cas9-mediated generation of light chain KI mice was rapid and reliable.

## One-step KI mice generation expressing genuine human germline BCRs by multiplexing H and L chain in zygotes

After establishing that we can individually rapidly generate H or L chain KI mice, we wanted to determine if we would be able to obtain the simultaneously insert pre-rearranged Ig H and L inserts into their respective native loci via CRISPR-mediated injection with two donor plasmids. The advantage of multiplexed HDR knock-in of H and L chains is that it will eliminate extra crosses between mice expressing H and L chains alone and save a considerable amount of time; however, the generation of double knock-in model is also a

challenge. Indeed, the insertion of two large constructs at precise and native loci by zygote microinjection has not been reported so far. In order to test this notion, we attempted to generate KI mice expressing paired CLK21 H and L chains, which constitute one of the native human germline VRC01-class BCRs identified from a HIV-1 seronegative volunteer (Havenar-Daughton et al, 2018b).

To expedite the generation of the CLK21 KI model expressing both the CLK21 germline H and L chain sequences, fertilized mouse oocytes were microinjected with two donor plasmids each bearing CLK21 gl H and κ pre-rearranged sequences and the relevant 5′ and 3′ homology arms, mouse VHJ558 or Vκ4-53 promoter and corresponding leader regions, and four sgRNAs—with two sgRNAs targeting each H or κ locus, and Cas9 (Appendix Fig S1C). After the implantation of the injected fertilized zygotes into pseudo-pregnant females, fourteen F0 pups were born (Fig 2A). TaqMan genotyping analysis revealed that of the fourteen F0 founders, five were positive for either the CLK21 HC (3; 21.4%) or LC (2; 14.3%), and four (28.6%) were positive for both HC and LC (Fig 2A and Appendix Table S3). Next, we crossed one of our positive heterozygote $H^{CLK21/WT}$ $\kappa^{CLK21/WT}$ F0 founders with WT mice to follow the frequency of germline transmission of the CLK21 KI H/L chains by genotyping. A total of 11 F1 pups were born, and although the numbers were small, out of the eleven F1 animals, four were $H^{CLK21/WT}$ $\kappa^{CLK21/WT}$ and five were $H^{WT/WT}$ $\kappa^{CLK21/WT}$, and two were $H^{WT/WT}$ $\kappa^{WT/WT}$ (Fig 2B) indicating that the CLK21 H and L alleles were transmitted to the progeny and appeared to segregate in a Mendelian ratio.

To determine whether the insertion of CLK21 germline H and L chain affected B-lymphocyte development, we compared the bone marrow progenitors of 8- to 10-week-old $H^{CLK21/WT}$ $\kappa^{CLK21/WT}$ (referred to hereafter as CLK21) mice versus WT control animals using the Hardy classification system (Hardy et al, 1991). The proportions of early (B220$^+$CD43$^+$) and late (B220$^+$CD43$^-$) B cells in the bone marrow of CLK21 and WT mice were similar as revealed by flow cytometry (Appendix Fig S2A). Further analysis of early (BP-1$^+$CD24$^+$) B cells showed comparable frequencies of these subfractions in the CLK21 and WT mice, while we observed a decrease in fraction F of late (IgD$^+$ IgM$^+$) B cells in CLK21 compared to WT mouse (Appendix Fig S2B and C). These results suggest that CLK21 KI B cells readily crossed the tolerance checkpoints in the bone marrow for further maturation in the spleen.

To further explore the maturation of CLK21 transitional B cells, we compared the splenic lymphocytes of CLK21 with those of WT mice. Consistent with our bone marrow results, we found similar frequencies of B and T cells in the spleens of CLK21 and WT mice

---

**Figure 2. Generation of CLK21 human BCR knock-in mouse model.**

A  Table shows the total number of pups, the number or frequency of human HC-, human LC-, and human BCR KI pups after One-step CRISPR/Cas9 microinjection of CLK21.

B  CLK21 KI mice F0 and F1 generations. Squares represent male mice and circles represent female mice. Upper halves of squares or circles represent Ig κ, and the lower halves represent Ig H, as shown in the schematic. F0 generation mice genotyping results showing 7, 9 and 14 are $H^{CLK21/WT}\kappa^{WT/WT}$, 1 and 4 are $H^{WT/WT}\kappa^{CLK21/WT}$, and 2, 3, 6, and 12 are $H^{CLK21/WT}\kappa^{CLK21/WT}$. Mouse 3 was crossed with WT to obtain 11 F1 progeny: Five F1 mice are $H^{WT/WT}\kappa^{CLK21/WT}$, four are $H^{CLK21/WT}\kappa^{CLK21/WT}$ and two are WT.

C  Single-cell sequencing for naïve B cells from heterozygous CLK21 double KI mice. Left column shows the gating strategy for sorting naïve B cells (upper) and eOD-GT8-specific naïve B cells (lower). Right pie charts show the frequency of paired HC and LC sequences among total naïve B cells (upper) and eOD-GT8-specific naïve B cells (lower).

D  Germline-targeting eOD-GT8 binding activity of B cells from WT, $H^{CLK21/WT}\kappa^{WT/WT}$, $H^{WT/WT}$ $\kappa^{CLK21/WT}$), and $H^{CLK21/WT}$, $\kappa^{CLK21/WT}$ KI mice. 8-week-old mice were detected by FACS. X and Y axes represent that BCR were stained with eOD-GT8 tetramer conjugated with Alexa Fluor™ 488 and Alexa Fluor™ 647, respectively. Representative dots were gated from Scatter/Singlet/Live (SSL), B220$^+$ IgM$^+$ IgD$^+$.

E  Quantification of eOD-GT8 binding in CLK21 KI mice. X-axis represents F0 ($n = 1$) and F1 ($n = 3$) KI animals, and the $Y$-axis represents the percentage of eOD-GT8 targeting binders in mature (IgM$^+$ IgD$^+$) B cells. Bars indicate mean ± SD from mice in each group.

A

| Line | # pups | # knock-in (KI) pups | | | KI pup frequency (%) | | |
|------|--------|------|------|------|------|------|------|
| | | *H* | *κ* | *H, κ* | *H* | *κ* | *H, κ* |
| **CLK21** | 14 | 3 | 2 | 4 | 21.4 | 14.3 | 28.6 |

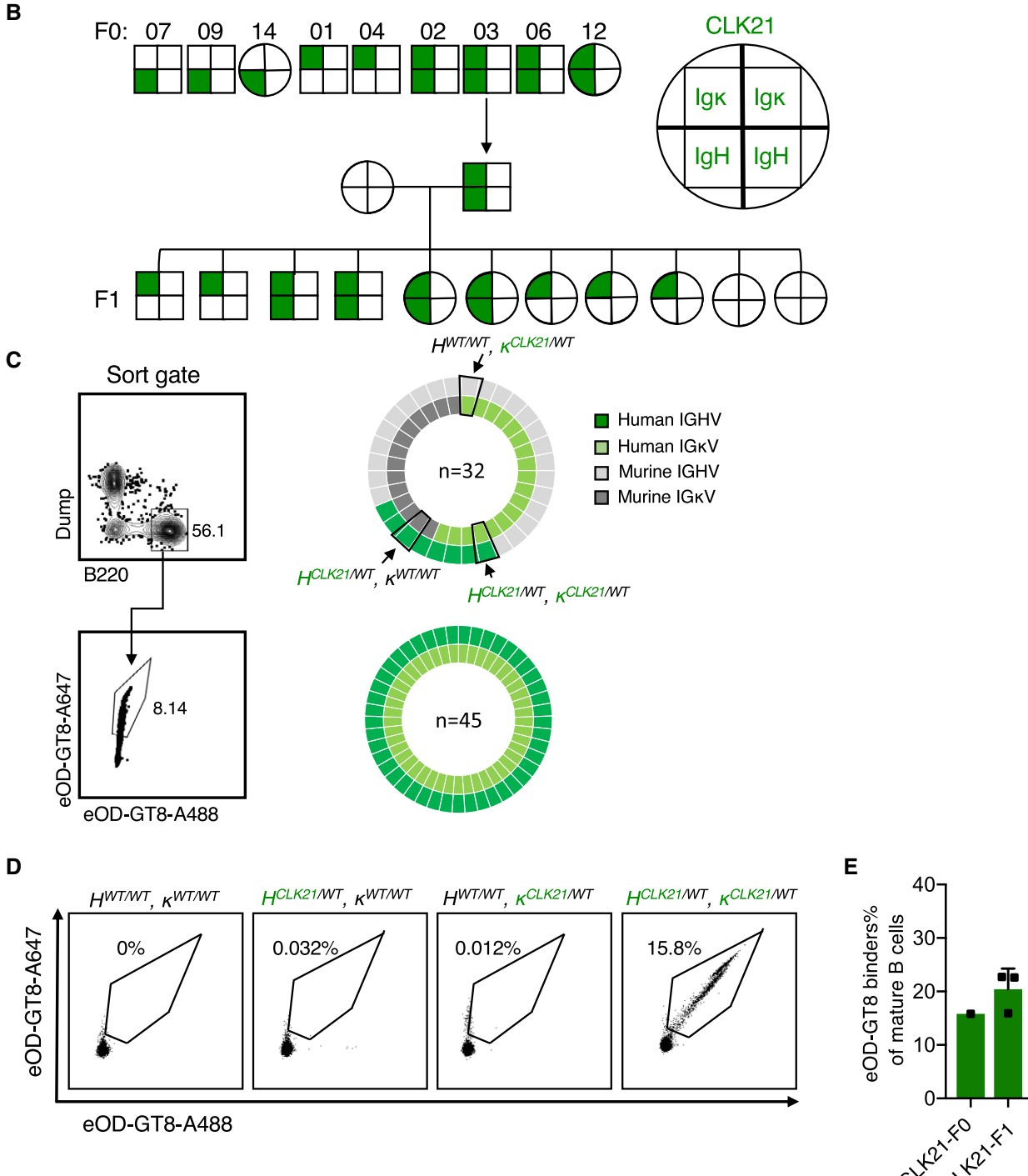

Figure 2.

(Appendix Fig S3A and B). Moreover, we observed similar number of mature follicular (CD21$^{hi}$CD24$^{lo}$), marginal zone B cells (CD21$^{hi}$CD24$^{hi}$CD23$^{-}$), T1 cells (CD21$^{lo}$CD24$^{hi}$) and T2 cells (CD21$^{hi}$CD24$^{hi}$CD23$^{+}$) (Appendix Fig S3C and D) in CLK21 and WT mice. Taken together, our results show that there was no significant change in the splenic lymphocyte population following CLK germ-line heavy and light chain insertion.

Upon observing no adverse impact on B-cell development in the CLK21 KI mice, we proceeded to profile the naïve B-cell repertoire. First, we detected the expression of HC and LC in mice with reverse transcription PCR (RT–PCR) and amplified cDNA using specific primers (Appendix Table S6). We found double expression of HC and LC is only from the CLK21 KI mice that were positive for the human BCRs (Appendix Fig S4A). Next, we generated BCR libraries from peripheral blood RNA isolation by next generation sequencing from WT, $H^{CLK21/WT}\kappa^{WT/WT}$, $H^{WT/WT}\kappa^{CLK21/WT}$, and $H^{CLK21/WT}\kappa^{CLK21/WT}$ mice, focusing on those mice expressing the highest levels of HC or LC or HC + LC according to RT–PCR. As expected, there were no sequences corresponding to CLK21 H or L chains in the WT mouse. In contrast, the clonotypes corresponding to H or L chains were present in the $H^{CLK21/WT}\kappa^{WT/WT}$ (44.6%) or $H^{WT/WT}, \kappa^{CLK21/WT}$ (7.9%) mouse. Finally, we observed that CLK21 HC and CLK21 LC made up 26.3% and 14.2%, respectively, of all the clonotypes detected in $H^{CLK21/WT}\kappa^{CLK21/WT}$ mouse (Appendix Fig S4B). Single B-cell repertoire sequencing of naïve B cells from a heterozygous CLK21 mouse line revealed that 12.5% (4/32) of sequenced BCRs were CLK21 HC + LC pairs; 12.5% were CLK21 HC paired with mouse LC; 43.8% (14/32) were CLK21 LC paired with mouse HC; and 31.3% (10/32) were mouse HC + LC pairs (Fig 2C). Sequencing of naïve B cells sorted by eOD-GT8 probes which target the VRC01-class bnAb precursors showed that 100% (45/45) of sequenced BCRs were CLK21 HC + LC pairs (Fig 2C). These results indicated that the CLK21 mouse line displays antigen-specific human BCRs *in vivo*. Having confirmed that we can indeed detect CLK21 BCR sequences, we proceeded to test whether BCRs expressing CLK21 germline HC and LC sequences were functional. To this end, we isolated peripheral B cells from $H^{CLK21/WT} \kappa^{WT/WT}$, $H^{WT/WT} \kappa^{CLK21/WT}$, and $H^{CLK21/WT}\kappa^{CLK21/WT}$ mice and examined their ability to bind biotinylated eOD-GT8 probe. B cells from WT animals ($H^{WT/WT}, \kappa^{WT/WT}$) were used as controls. We found that 15.8% of the naïve B cells from the $H^{CLK21/WT} \kappa^{CLK21/WT}$ mouse were able to bind this probe, whereas only 0.032, 0.012, and 0% of naïve B cells from a $H^{CLK21/WT}\kappa^{WT/WT}$, $H^{WT/WT}\kappa^{CLK21/WT}$ and WT mouse, respectively, were able to bind the eOD-GT8 probe (Fig 2D and E). These results validate the requirement of both CLK21 H and L chains for eOD-GT8 immunogen binding. Furthermore, consistent with the absence of IgHV1-2 in the mouse B-cell repertoire, and the specificity of eOD-GT8 for its cognate antibody, we found that WT B cells are incapable of binding streptavidin-conjugated eOD-GT8 (Fig 2D). Taken collectively, we have shown that the CLK21 double KI mouse represents a valuable model for examining eOD-GT8-specific immunoresponses.

To convince ourselves of the value of a CRISPR/Cas9-mediated multiplexed HDR template strategy in generating KI mice expressing predetermined pairs of H and L chains, we examined whether this strategy would work for H + L chain pairs of the other VRC01 bnAb class. Therefore, whereas CLK21 utilizes Vκ1-5, we tested this approach with two VRC01-class, N6 sub-class germline BCRs

utilizing Vκ1-33 but also capable of eOD-GT8 binding, CLK09 and CLK19 (Havenar-Daughton *et al*, 2018b). The affinities of these BCRs for eOD-GT8 range from relatively high affinities of 350 nM (CLK09) and 440 nM (CLK21), to a moderate affinity of 1.8 μM (CLK19), close to the geomean affinity of eOD-GT8 for human naive VRC01-class precursors (3.0 μM) (Abbott *et al*, 2018; Havenar-Daughton *et al*, 2018b). Using the strategy described above, double injections with the CLK09 (Ig H + Ig κ) or CLK19 (Ig H + Ig κ) constructs resulted in 10 or 30 F0 founders, respectively (Appendix Fig S1C and Fig 3A). TaqMan genotyping analysis of CLK09 revealed that of the 10 F0 founders, one was positive for the HC only (10%) and two were positive for the HC + LC (20%) (Fig 3A and Appendix Table S4). In the case of CLK19, 12/30 (40%) of the F0 founders had H, L, or H + L chain insertions: seven were positive for the CLK19 HC only (23.3%) and two were positive for light chain only (6.6%); three (10%) were positive for both HC and LC (Fig 3A and Appendix Table S5). RT–PCR results with specific primers revealed that only double KI CLK09 or CLK19 mice produced high levels of both HC and LC RNA (Appendix Fig S5A and C) Crossing one of our positive $H^{CLK09/WT} \kappa^{CLK09/WT}$ (referred to hereafter as CLK09) F0 founders with a WT mouse to determine the frequency of germline transmission resulted in four F1 progeny: one $H^{CLK09/WT} \kappa^{WT/WT}$, one $H^{WT/WT} \kappa^{CLK09/WT}$, and two WT mice with no CLK09 insertions (Appendix Fig S5B). After crossing the positive F0 $H^{CLK09/WT} \kappa^{CLK09/WT}$ with a F1 $H^{CLK09/WT} \kappa^{WT/WT}$, we obtained both heterozygous and HC homozygous mice ($H^{CLK09/WT} \kappa^{CLK09/WT}$ and $H^{CLK09/CLK09} \kappa^{CLK09/WT}$) (Appendix Fig S5B). Similarly, we crossed one of our positive $H^{CLK19/WT}\kappa^{CLK19/WT}$ (hereafter referred to as CLK19) F0 founders with a WT mouse and obtained a total of five F1 progeny. Genotyping results showed that two of the F1 mice were $H^{CLK19/WT}\kappa^{CLK19/WT}$, one mouse was $H^{CLK19/WT} \kappa^{WT/WT}$ and two were $H^{WT/WT}\kappa^{CLK19/WT}$ (Appendix Fig S5D). In both cases, our genotyping data indicate that the KI alleles appear to exhibit Mendelian inheritance.

We analyzed B-cell development in CLK09 and CLK19 KI mice and found that their bone marrow and splenic B-cell subpopulations were normal (Appendix Figs S2A–C and S3A–D), suggesting no major impact of CLK09 or CLK19 germline H and κ chain insertions. To ascertain the functionality of B cells expressing CLK09 and CLK19 BCRs, we measured the eOD-GT8 binding ability of peripheral B cells from heterozygous CLK09 and CLK19 as well as a HC homozygous F2 CLK09 mouse. We found that 14.9 and 23.2% of the naïve B cells from CLK09 and CLK19 mice and 62.7% of the naïve B cells from CLK09 HC homozygous mouse, respectively, were able to bind strep-tavidin-conjugated eOD-GT8 (Fig 3B and D). Furthermore, the ability to bind streptavidin-conjugated eOD-GT8 was maintained in an F2 generation CLK09 mouse as well as an F1 generation CLK19 animal (Fig 3C and E). All together, our results reveal that our CRISPR/Cas9-mediated multiplexed HDR template strategy is capable of rapidly generating mouse models requiring the expression of both human Ig H and L chains for validating immunogens for vaccine development.

## B cells bearing native human BCR specificities can be primed and recruited to GCs by eOD-GT8 60mer even at rare precursor frequencies

Since the three KI mouse lines (CLK21, CLK19, and CLK09) all bear functional BCRs that are proficient in eOD-GT8 tetramer binding, we

**A**

| Line | # pups | # knock-in (KI) pups | | | KI pup frequency (%) | | |
|---|---|---|---|---|---|---|---|
| | | *H* | *κ* | *H, κ* | *H* | *κ* | *H, κ* |
| **CLK09** | 10 | 1 | 0 | 2 | 10 | 0 | 20 |
| **CLK19** | 30 | 7 | 2 | 3 | 23.3 | 6.6 | 10 |

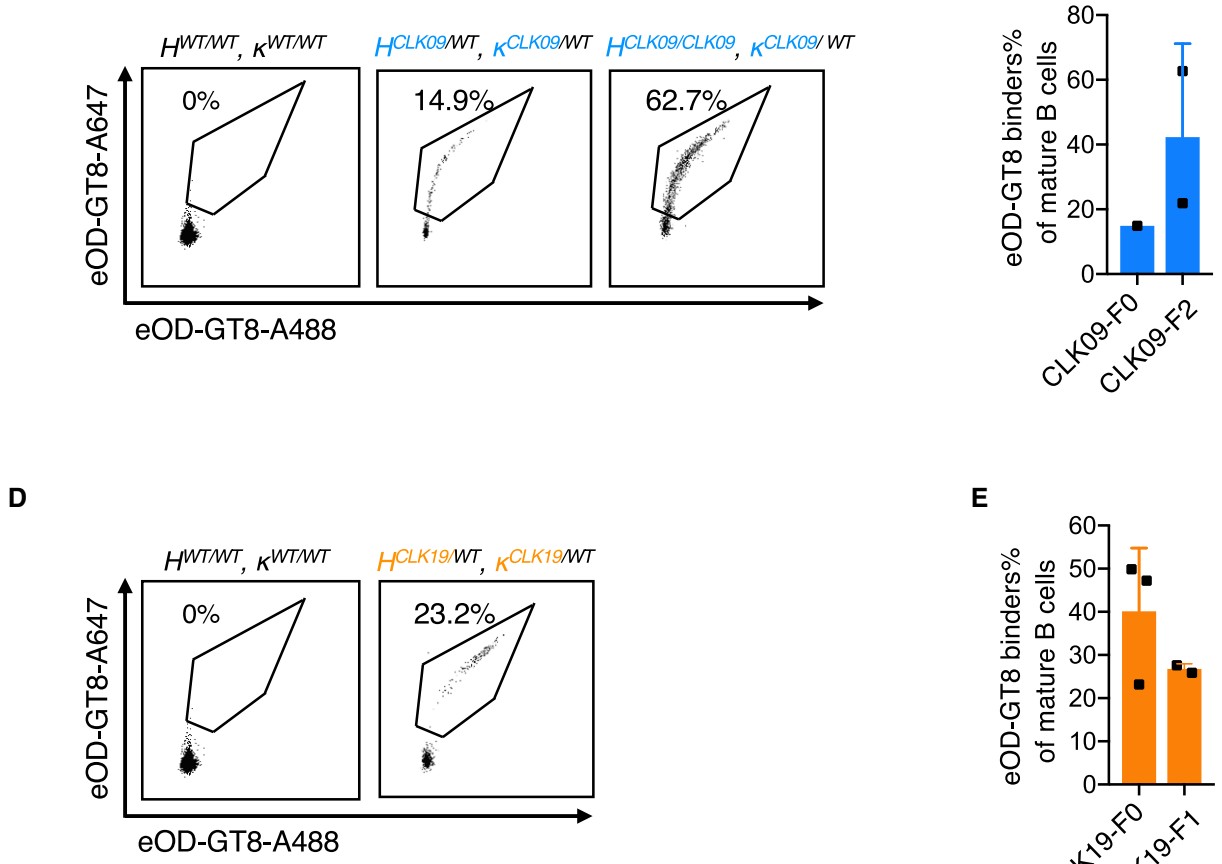

**Figure 3. Generation of CLK09 and CLK19 human BCR knock-in mouse model.**

A   Table shows the total number of pups, the number or frequency of human HC-, human LC-and human BCR KI pups after One-step CRISPR/Cas9 microinjection of CLK09 and CLK19.

B   Binding activity detection of heterozygous CLK09 human BCR KI mice to eOD-GT8.8-week-old mice were detected by FACS. *X*-axis and *Y*-axis represent that BCR were stained with eOD-GT8 tetramer conjugated with Alexa Fluor™ 488 and Alexa Fluor™ 647, respectively. Representative dots were gated as in Fig 2D.

C   Quantification of eOD-GT8 targeting CLK09 KI mice. *X*-axis represents F0 (*n* = 1) and F2 (*n* = 2) pups, *Y*-axis represents as in Fig 2E. Bars indicate mean ± SD from mice in each group.

D   Binding activity detection of heterozygous and homozygous CLK19 human BCR KI mice to eOD-GT8. 8-week-old mice were detected by FACS. *X* and *Y* axes represent as in (B).

E   Quantification of eOD-GT8 binding in $H^{CLK19/WT} \kappa^{CLK19/WT}$ KI mice. *X*-axis represents F0 (*n* = 3) and F1 (*n* = 2) animals, *Y*-axis represents as in Fig 2E. Bars indicate mean ± SD from mice in each group.

wanted to determine whether these B-cell precursors could be triggered *in vivo* by the eOD-GT8 60mer immunogen currently in a Phase I clinical trial (https://clinicaltrials.gov/ct2/show/NCT03547245). This is of particular importance given that these are not inferred BCR sequences, but genuine human BCRs (Havenar-Daughton *et al*, 2018b). Because of the high precursor number observed in our KI lines, we decided to lower this precursor frequency by initially adoptively transferring 500,000 B cells from these KI mice, which express CD45.2 into congenic CD45.1 WT mice. One day after adoptive transfer (Day 0), we performed FACS staining to establish the number of specific eOD-GT8 CD45.2 B cells in the spleen. This measurement revealed a frequency of 1 in $10^4$ splenic CD45.2 cells (Fig 5A). Cohorts of recipient mice were immunized with eOD-GT8 60mer nanoparticles in Alhydrogel at Day 0, and their spleens were harvested for measuring GC formation (CD38$^{low}$CD95$^+$) at Day 8. At this time point, we observed that in all three lines, around 1% of the B cells were CD38$^{low}$CD95$^+$, demonstrating a robust GC formation (Fig 4A, first column FACS plots). We then used CD45.2 as a marker to interrogate the percentage of the GC B cells represented by our adoptively transferred CLK B cells. We observed that CLK19, CLK09, and CLK21 B cells accounted for ~ 3, ~ 9 and ~ 21% of the GC response, respectively, indicating that these cells were specifically recruited to the GC (Fig 4A, second column FACS plots). Importantly, in each case over 75% of the CD45.2 GC B cells were capable of binding the eOD-GT8 nanoparticles, but not its epitope inactivated version (eOD-GT8-KO), consistent with the expression of the various VRC01-class precursors on their surface (Fig 4A, last two columns FACS plots). The FACS data have been quantified and are shown in Fig 4B. In addition, we also verified the presence of adoptively transferred CLK09 and CLK19 B cells within GCs by immunofluorescence imaging of splenic sections which is consistent with the flow cytometry results (Fig 4C). Taken together, our results show that naïve B cells expressing native human BCRs can be activated *in vivo* by eOD-GT8 60mer and recruited to GCs.

In humans, it has been determined that the frequency of eOD-GT8-specific VRC01-class naïve B-cell precursors is approximately 1 in 400,000 (Havenar-Daughton *et al*, 2018b). Therefore, we wondered whether naive B cells expressing genuine human BCRs could be activated by eOD-GT8 60mer at such low precursor frequencies. To test this notion, we again made use of the congenic adoptive transfer system, in which CD45.2 binder frequencies of CLK19, CLK09, and CLK21 specificities were present at 1 in $10^4$, 2 in $10^5$ or 3 in $10^6$ at the time of eOD-GT8 60mer immunization (Fig 5A; Abbott *et al*, 2018; Steichen *et al*, 2019). Cohorts of these mice were immunized with eOD-GT8 60mer in Alhydrogel 24 h after adoptive transfer. At Day 8 post-immunization, we observed that under the similar formation of GC (~ 1%) among B cells in CLK adoptively transferred mice with different precursor (Appendix Fig S6A), the percentage of CD45.2 cells recruited to the GC was dependent on precursor frequency. In addition, these CD45.2 cells could also be found even at 3 in $10^6$, which is in the low range of the physiological frequency found in humans (Fig 5B and C). Of note, in each case the majority of the CD45.2 B cells were specific for eOD-GT8 nanoparticles, rather than eOD-GT8KO, even though eOD-GT8KO and eOD-GT8 induced similar GC formation in CLK adoptively transferred mice with different precursor frequency (Fig 5B and C, and Appendix Fig S6B), which indicates the expected epitope specificity. Taken together, our results show that in response to eOD-GT8 60mer, B cells bearing CLK19, CLK09, and CLK21 BCRs can be recruited to murine GCs when initially present at a human physiological frequency.

## B cells expressing CLK19, CLK09, and CLK21 BCRs exhibit different GC kinetics after eOD-GT8 60mer immunization *in vivo*

Next, we wanted to examine how eOD-GT8 60mer immunization affects the kinetics of GC response, as well as the accumulation of somatic hypermutations, after the adoptive transfer of CLK19, CLK09, and CLK21 native human B cells in a time course experiment. The results of these experiments will be extremely informative about the capacity of eOD-GT8 60mer immunization to trigger the longitudinal affinity maturation of these human antibodies. Accordingly, as described earlier, we adoptively transferred 1 in $10^4$ CD45.2 B cells from the CLK19, CLK09 or CLK21 donors into cohorts of CD45.1 recipients and immunized them with eOD-GT8 60mer and quantified the frequency of GC B cells on Day 8, 15, and 36. Our results show that in response to eOD-GT8 60mer, CLK21, CLK19, and CLK09 B cells were all recruited to the GC over time (Fig 6A). Furthermore, we observed that at Day 8 the number of CD45.2 B cells was relatively low for CLK19, probably reflecting the diminished affinity of this BCR for eOD-GT8. Interestingly, numbers of CLK19 B cells increased > 3-fold on Day 15 and they persisted on Day 36 (Fig 6A and B). By immunofluorescence imaging of splenic sections, we also confirmed the continued presence of CD45.2 CLK19 B cells within the GCs at Day 36 post-immunization, which

**Figure 4. GC responses on Day 8 in adoptively transferred mice with rare precursor frequency.**

A Germinal center (GC) response induced by eOD-GT8 60mer immunization in WT mice adoptively transferred with CD45.2 B cells from CLK21, CLK09, CLK19 mice on Day 8. 8-week-old CD45.1 mice were transferred with $5 \times 10^5$ (1 in $10^4$) of isolated CD45.2 B cells from CLK21, CLK09 and CLK19 KI one day before the immunization with eOD-GT8 60mer, on day 8, the splenocytes were isolated and detected by FACS. First column shows the frequency of total GC (CD38$^{lo}$CD95$^+$) among B cells gated from SSL, the second column shows the frequency of CD45.2 B cells among total GC, third column shows the frequency of eOD-GT8 binding B cells among GC CD45.2 B cells; fourth column shows the frequency of eOD-GT8 binding B cells that do not bind the VRC01 epitope-knockout, eOD-GT8-KO; diagrams at right show eOD-GT8 $K_D$ affinities for CLK21 (green, affinity 440 nM), CLK09 (blue, affinity 350 nM) and CLK19 (orange, affinity 1.8 μM).

B Quantification of GCs, CD45.2 cells and CD45.2 binder frequency in adoptively transferred mice. *X*-axis represents CLK21, CLK09 and CLK19 adoptively transferred mice group that immunized with eOD-GT8 60mer. *Y*-axis represents the frequency of total GC (left) among B cells, the frequency of CD45.2 cells within GCs and the frequency of eOD-GT8-CD45.2 binders. Each circle represents one mouse. *n* = 5 mice/group for CLK21 and CLK09, *n* = 3 mice/group for CLK19. Bars indicate geometric mean and geometric SD from mice in each group.

C Immunohistochemistry of Day 8 spleen sections for host mouse receiving $5 \times 10^5$ isolated CD45.2 B cells from CLK09 (left) and CLK19 (right) KI mouse. Green, B220; Blue, CD3; White, CD45.2; Red, GL7. White arrows indicate the CD45.2 B cells.

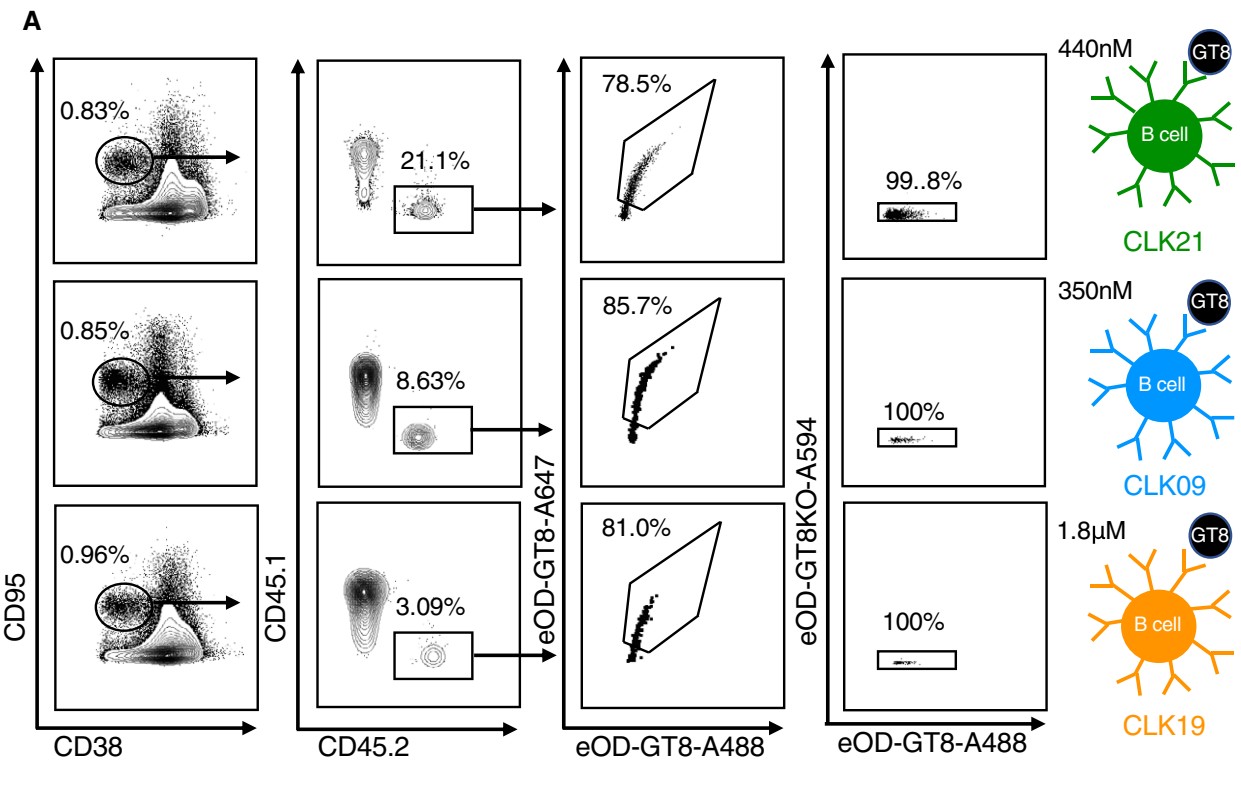

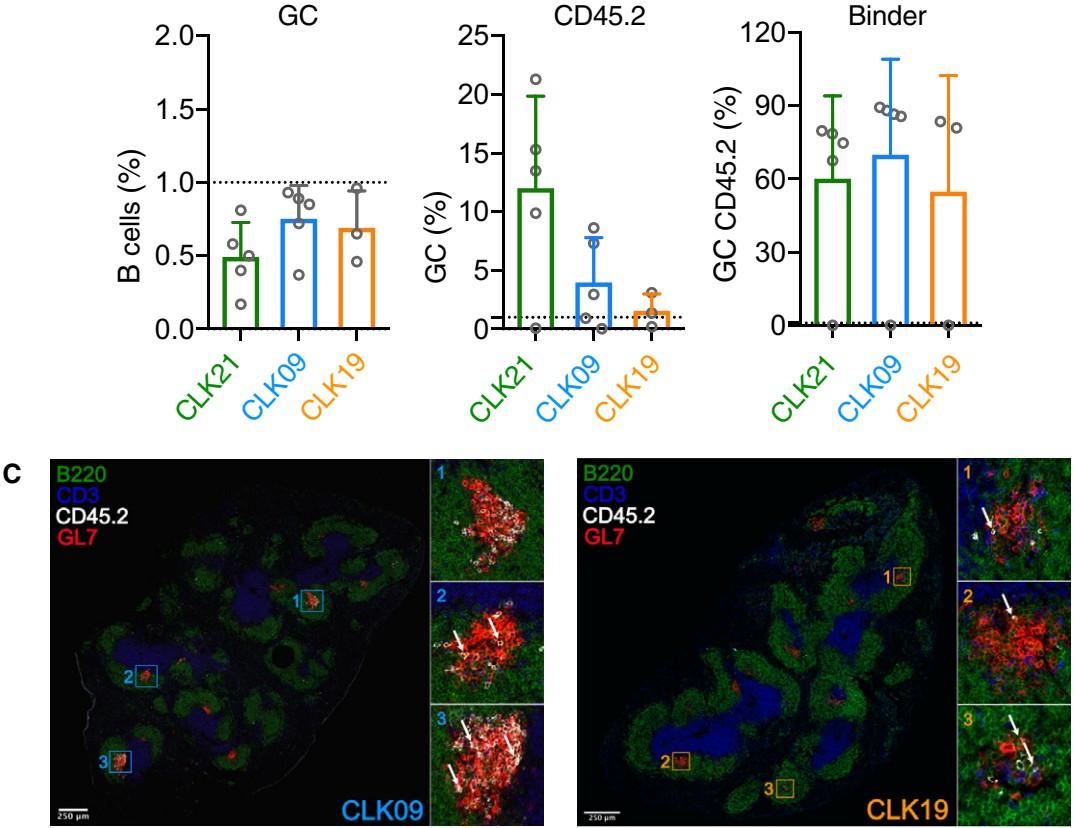

**Figure 4.**

is consistent with the flow cytometry results (Appendix Fig S7A). On the other hand, in the case of CLK09 and CLK21, on Day 8 the numbers of CD45.2 cells were between 7- to 12-fold higher than in the case of CLK19 adoptively transferred cells. Similar to what we observed in the case of CLK19, the CD45.2 B-cell numbers of CLK09 and CLK21 were amplified on Day 15 (Fig 6A and B). This may be reflective of the higher affinities of these two BCRs for eOD-GT8 compared with CLK19. Intriguingly however, for CLK09 and CLK21, although the percentage of GC B cells continued to be similar between Day 8 and 15, their numbers declined by Day 36; there was an over 10-fold decrease in their numbers at Day 36 when compared to Day 8 (Fig 6A). However, we could still observe some CD45.2 CLK09 B cells persisting in GC at Day 36 by immunofluorescence imaging (Appendix Fig S7B).

We expanded this experiment to include immunization at lower precursor frequencies. After the adoptive transfer of 2 in $10^5$ or 3 in $10^6$ CLK19, CLK09 and CLK21 B cells, we found that although BCRs for each of these VRC01-class BCRs can be activated, they exhibit precursor frequency-dependent CD45.2 GC kinetics (Fig 6B). Qualitatively consistent with the GC data, serum-binding analyses revealed that eOD-GT8 60mer induced significant epitope-specific IgG responses in CLK21, CLK19 and CLK09 B-cell recipients (Fig 6C). This demonstrated that activation of rare VRC01 precursor B cells led to serum Ab responses. In addition, we also assessed the frequency of class-switched CD45.2 IgG1$^+$ B cells in GC on Day 36 and observed that class-switched CD45.2 binders could be generated in most of the adoptively transferred CLK mice (Appendix Fig S7C and D), which is consistent with the ELISA results. Interestingly, the class-switched CLK CD45.2 B cells could also develop into the eOD-GT8-specific memory B cells (B220$^+$IgDlo CD38$^+$GL7$^-$CD45.2$^+$eOD-GT8$^+$IgM$^-$) (Appendix Fig S8A–C). Overall, these data reveal that in response to eOD-GT8, CLK19, CLK09, and CLK21 B cells can persist in the GCs, class switch, secrete antibody, and differentiate to memory B cells.

### B cells bearing VRC01-class human precursor BCRs undergo SHM and exhibit VRC01-class mutations after a single priming injection with eOD-GT8 60mer

To determine if B cells bearing VRC01-class BCRs undergo SHM and can accumulate VRC01-class mutations after immunization with eOD-GT8 60mer, we single-cell sorted eOD-GT8-specific and class-switched CLK B cells at different time points from adoptively transferred mice with 1 in $10^4$ of precursor frequency (Appendix Fig S9A) and performed single-cell PCRs to generate heavy-light chain sequence pairs for the mutation analysis (Lin *et al*, 2018). Notably, phylogenetic trees, for which HC-LC sequences were bioinformatically stitched together, revealed substantial increase in sequence diversification with time (Appendix Figs S9B and S9D, and Fig 7A). Further, there was a significant level of SHM with a median of 4 amino acid mutations in the human VH1-2 HC, common to our CLK19, CLK09, and CLK21 models (Fig 7B). While SHM levels were nearly identical in the VH1-2 HCs, the corresponding LCs of CLK19 (Vk1-33$^+$), CLK09 (Vk1-33$^+$) and CLK21 (Vk1-5$^+$) accumulated varying degrees of mutations ranging from a median of 1–3 aa mutations per LCs (Fig 7B). Of note, comparable SHM and sequence diversity also occurred in adoptively CLK09 and CLK21 transferred mice with 3 in $10^6$ of physiological precursor frequency (Appendix Fig S10A and B). Next, ratios of non-silent to silent mutations were assessed to further determine the impact of eOD-GT8 60mer immunizations. In VH1-2 HC, non-silent mutations occurred at a mean frequency of $N = 4.6 \pm 2.1$ nucleotides per V gene, surpassing the level of silent HC mutations (mean of $N = 1.8 \pm 1.2$ nucleotide mutations). The corresponding LCs exhibited mean non-silent mutation frequencies of $N = 1.5 \pm 1.6$ (CLK19) to approximately $N = 3.1 \pm 1.9$ nucleotides (CLK09 and CLK21) per V gene. These non-silent mutations were detectable in the CDR and FR of antibodies (Appendix Fig S9E).

SHMs were detected in HCDR1, HCDR2, and HCDR3 of the VH1-2 HC as well as LCDR1, LCDR2 and LCDR3 of the Vk1-33 and Vk1-5. Interestingly, the accrual of HCDR3 mutations was more prominent in the CLK19 and CLK09 models when compared to CLK21 and LCDR3 mutations also occurred in CLK19 and CLK21 (Fig 7C). Further, we observed that the 5-aa motif (CCQQYXTF) in LCDR3, which is a natural feature of our three genuine CLK lines, was preserved following eOD-GT8 60mer immunization (Fig 7C, WEBLOGOS). To further delineate eOD-GT8 60mer-induced B-cell responses, we assessed the occurrence of VRC01-class mutations in all available VH1-2 HCs (irrespective of pairing) to evaluate whether a single eOD-GT8 60mer priming could initiate a desired affinity maturation pathway toward mature VRC01-class bnAbs (Fig 7D, Appendix Fig S9C; Briney *et al*, 2016; Abbott *et al*, 2018). Thirty-six days after single eOD-GT8 60mer priming we observed substantial numbers of VRC01-class mutations (Fig 7D). The similar evolution

**Figure 5. GC responses on Day 8 in adoptively transferred mice with different precursor frequency.**

A Precursor frequency in CLK21, CLK09 and CLK19 adoptively transferred mice. 8-week-old CD45.1 mice were adoptively transferred with $5 \times 10^5$, $1 \times 10^5$ or $2 \times 10^4$ CD45.2 B cells from CLK21, CLK09 or CLK19 KI mice 1 day before immunization with eOD-GT8 60mer. 12 h post-adopt transfer, splenocytes were isolated and detected by FACS. X-axis represents the cell number of transferred CD45.2 B cells, y-axis represents the precursor frequency which is defined as (Number of CD45.2 binders)/(Number of B cells) for three lines. Each circle represents each mouse in each group. $n = 2$–4 mice/group for CLK21 and CLK09; $n = 3$ mice/group for CLK19. Bars indicate mean $\pm$ SD from mice in each group.

B CD45.2 frequency within GCs on Day 8 in CLK adoptively transferred mice with different precursor frequency. Adoptively transferred mice with different precursor frequency were immunized with eOD-GT8 60mer, on Day 8, the splenocytes were isolated and the CD45.2 populations were detected by FACS with the marker SSL$^+$B220$^+$CD95$^+$CD38$^-$CD45.2$^+$. Gated plots represent the CD45.2 frequency within GCs.

C Quantification of the frequency of CD45.2 and CD45.2 binders. Upper graph shows the quantification of CD45.2 frequency among GC in three adoptively transferred mouse models after the immunization of eOD-GT8 60mer. The graphs below show the frequency of GC CD45.2 binders. X-axis represents the different precursor frequency group. Y-axis represents the percentage of GC (upper) and GC CD45.2 B cells (lower), respectively. Each circle represents one mouse. Mice from two independently repeated experiments were analyzed, $n = 5$–10 mice/group. Bars indicate geometric mean and geometric SD from mice in each group. Significant differences were calculated with Student's *t*-test and shown as: $P > 0.05$, no statistical significance (ns), $*P < 0.05$, $**P < 0.01$, $***P < 0.001$, $****P < 0.0001$. All $P$ value analyses were calculated by GraphPad Prism V8.0.

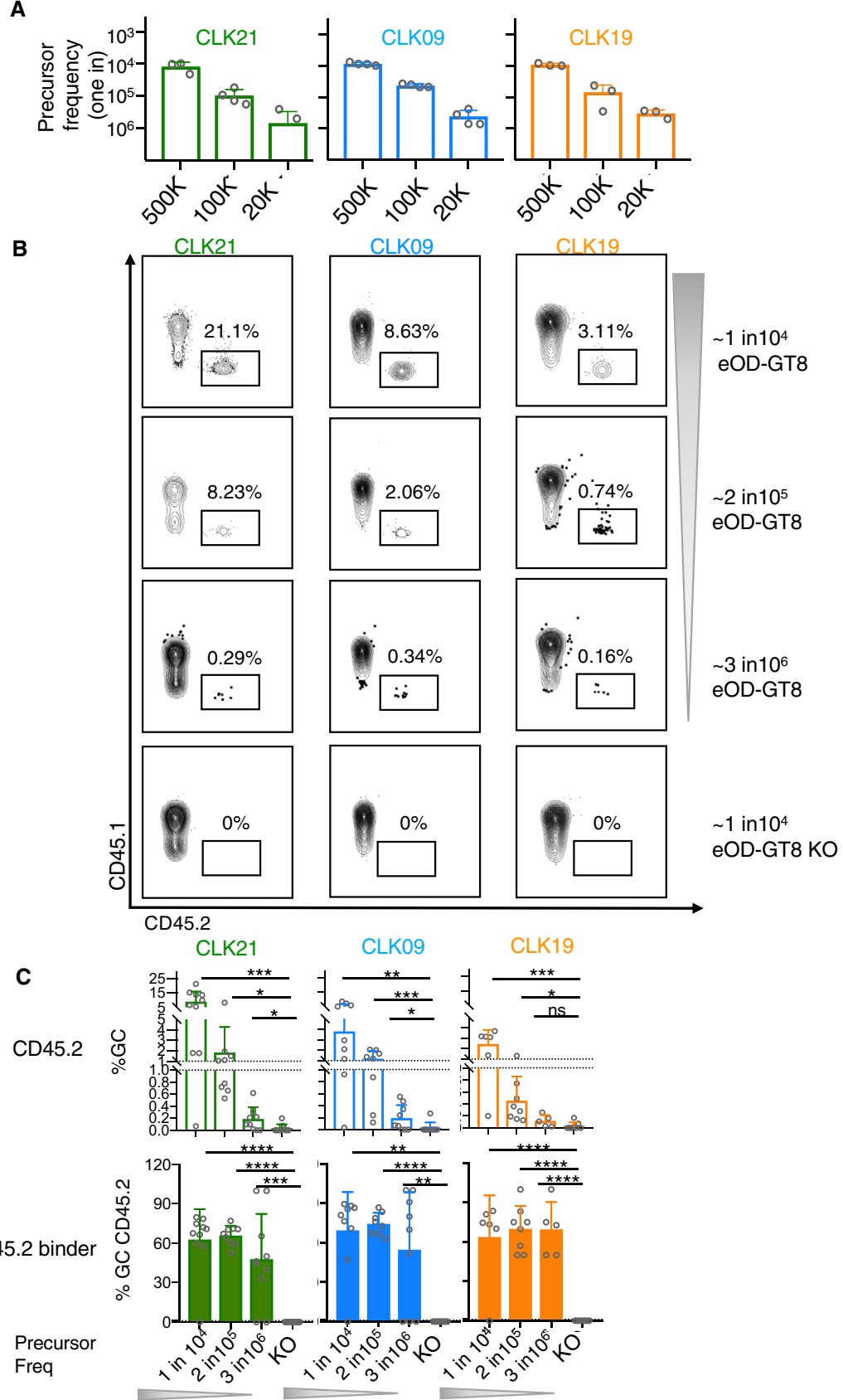

**Figure 5.**

also occurred in adoptively CLK09 and CLK21 transferred mice with 3 in $10^6$ physiological precursor frequency (Appendix Fig S10C and D). In addition, the VH1-2 positions M34 and K63, reported to be important positions for the development of VRC01-class mutations (Jardine *et al*, 2016b), were in fact mutated at reasonable frequency. M34L/I change were observed for all three CLK lines, whereas Q63L/R changes appeared to be more pronounced in lines CLK19 and CLK21 (Appendix Fig S9F).

Here, we observed that B cells naturally occurring in the human population can undergo significant SHM and accumulate bnAb-like mutations after a single priming immunization. These results support the feasibility of priming rare precursor B cells in the ongoing eOD-GT8 60mer clinical study.

## Discussion

The importance of genetically engineered mouse models in basic and pre-clinical research is indisputable. Here, we are demonstrating the feasibility of co-injecting fertilized zygotes with Cas9 and two donor plasmids, one for each of the H and L chains, capable of simultaneously targeting the murine Ig H and κ chain native genomic loci.

The generation of mice bearing pre-rearranged H or L chain antibody sequences has made it possible to address fundamental questions about B-cell development and determine how these cells are dynamically altered/differentiated in response to antigenic challenges (Weaver *et al*, 1985; Nussenzweig *et al*, 1987; Manz *et al*, 1988; Goodnow *et al*, 1989; Bloom *et al*, 1993; Benschop *et al*, 2001; Jardine *et al*, 2015). Based on the reported work on CRISPR/Cas9-mediated HDR in zygotes (Wang *et al*, 2013; Yang *et al*, 2013), we previously generated two mouse models bearing 1.9 kb pre-arranged human Ig PGT121$^{gH}$ or BG18$^{gH}$ precursors at the endogenous mouse Ig H locus (Lin *et al*, 2018; Steichen *et al*, 2019). Our fast-track KI methodology for rapidly introducing human antibody heavy chains into the actual Ig H locus played a key role during the evaluation of germline-targeting strategy for a HCDR3-dominant antibody (Steichen *et al*, 2019). However, employing CRISPR/Cas9-mediated knock-in of large fragments at loci other than ROSA26 (Chu *et al*, 2016) and inserting human light chains into the endogenous mouse Ig κ locus of fertilized zygotes still presents a major obstacle to researchers (Jacobsen *et al*, 2018).

Previous examples of LC insertions at the κ locus have either relied on ES cells (Pelanda *et al*, 1997) or when zygote microinjections using CRISPR/Cas9 have been performed, this has been of bicistronic Ig H and L chains targeted to the Ig H locus (Jacobsen *et al*, 2018). In light of this, we believe that our strategy of co-injection of fertilized zygotes

with Cas9 and two donor plasmids, one for each of the H and L chains, capable of simultaneously targeting the murine Ig H and κ chain genomic loci, constitutes a technological leap. While strategies for multiple HDR-directed targeted insertions have been shown to be possible *ex vivo* in cell lines (Shin *et al*, 2015; Tran *et al*, 2019), to the best of our knowledge, ours is the first demonstration of *in vivo* multiple insertions. This accomplishment considerably enhances the CRISPR/Cas9-mediated genome modification toolkit. Furthermore, unlike F0 animals generated from ES cells, which require numerous backcrosses in order to achieve the complete germline transmission of insertions or deletions, F0 progeny obtained after zygote infections do not require extensive crossings to obtain a stable line. Thus, our methodology significantly reduces the time to obtain a mouse line that can enter a pre-clinical pipeline.

Our work also shows for the first time the generation and immunocharacterization of KI mice bearing human VRC01-class BCR sequences (Figs 2 and 3, and Appendix Fig S5). This is in contrast to prior studies, which have relied on germline-reverted antibody sequences for generating KI mouse models for pre-clinical vaccine development (Dosenovic *et al*, 2015; Jardine *et al*, 2015; Briney *et al*, 2016; Escolano *et al*, 2016; Abbott *et al*, 2018; Saunders *et al*, 2019; Steichen *et al*, 2019). The drawback to using inferred germline antibody sequences for generating KI mice is that they are bioinformatically derived from mature bnAbs or in some cases from longitudinal sequencing data on bnAb lineages and therefore may not be comparable to bona fide human naïve BCRs in immunogen binding and priming responses. Thus, in this study, we generated three KI mouse lines: CLK21, CLK09, and CLK19, concomitantly expressing H and L VRC01-class BCRs (Figs 2 and 3, and Appendix Fig S5). All three BCRs were previously identified from B cells of HIV-1 seronegative volunteers via eOD-GT8-binding (Havenar-Daughton *et al*, 2018b). Epitope-specific precursor sorting and high-throughput sequencing now allows us to identify BCR precursors (Jardine *et al*, 2016a; Havenar-Daughton *et al*, 2018b; Steichen *et al*, 2019). We found that B cells bearing pre-rearranged germline human H and L chain variable regions respond to germline-targeting immunogens in the same manner as iGLs: such B cells are recruited to germinal centers (Fig 4–6), secrete class-switched antibodies (Fig 6C), undergo somatic hypermutation (Fig 7, and Appendix Figs S9 and S10), and differentiate into memory B cells (Appendix Fig S8). We believe that the technology reported here to rapidly generate H + L paired KI mice promises to play an important role in future immunogen validation and iterative optimization.

Of direct relevance to HIV vaccine development, we generated three novel VRC01-class mouse models for stringent pre-clinical evaluation of the germline-targeting immunogen eOD-GT8 60mer.

**Figure 6. Duration of GC response in CD45.2 adoptively transferred mice.**

A   CD45.2 frequency among GCs induced by eOD-GT8 60mer at different time points. Adoptively transferred mice with 1 in $10^4$ of precursor frequency were immunized with eOD-GT8 60mer. On days 8, 15 and 36, the splenic B cells were isolated to measure the CD45.2 cell frequency in GC B cells by FACS using the markers as in Fig 5B mentioned above by FACS. Gated plots represent the CD45.2 frequency among GC B cells.

B   Quantification of CD45.2 frequency among GCs at three time points. Adoptively transferred mice groups with different precursor frequencies (1 in $10^4$, 2 in $10^5$, 3 in $10^6$) at three time points, Y-axis represents CD45.2 frequency among GC. Each circle represents one mouse. For CLK21 and CLK09 panel, $n = 5$ mice in each group. For CLK19 panel, $n = 3$ mice in each group. Bars indicate mean ± SD from mice in each group.

C   IgG titers detected by ELISA. eOD-GT8 or eOD-GT8 KO soluble proteins were used to coat ELISA plates. Sera from adoptively transferred mice at different time points were used to detect by ELISA. Bars indicate geometric mean and geometric SD from mice in each group. For CLK21 and CLK09 panel, $n = 5$ mice in each group. For CLK19 panel, $n = 3$ mice in each group. X-axis represents precursor frequency, and Y-axis represents the change of area under curve ($AUC_{coated\ eOD-GT8} - AUC_{coated\ eOD-GT8\ KO}$). Each circle represents one mouse.

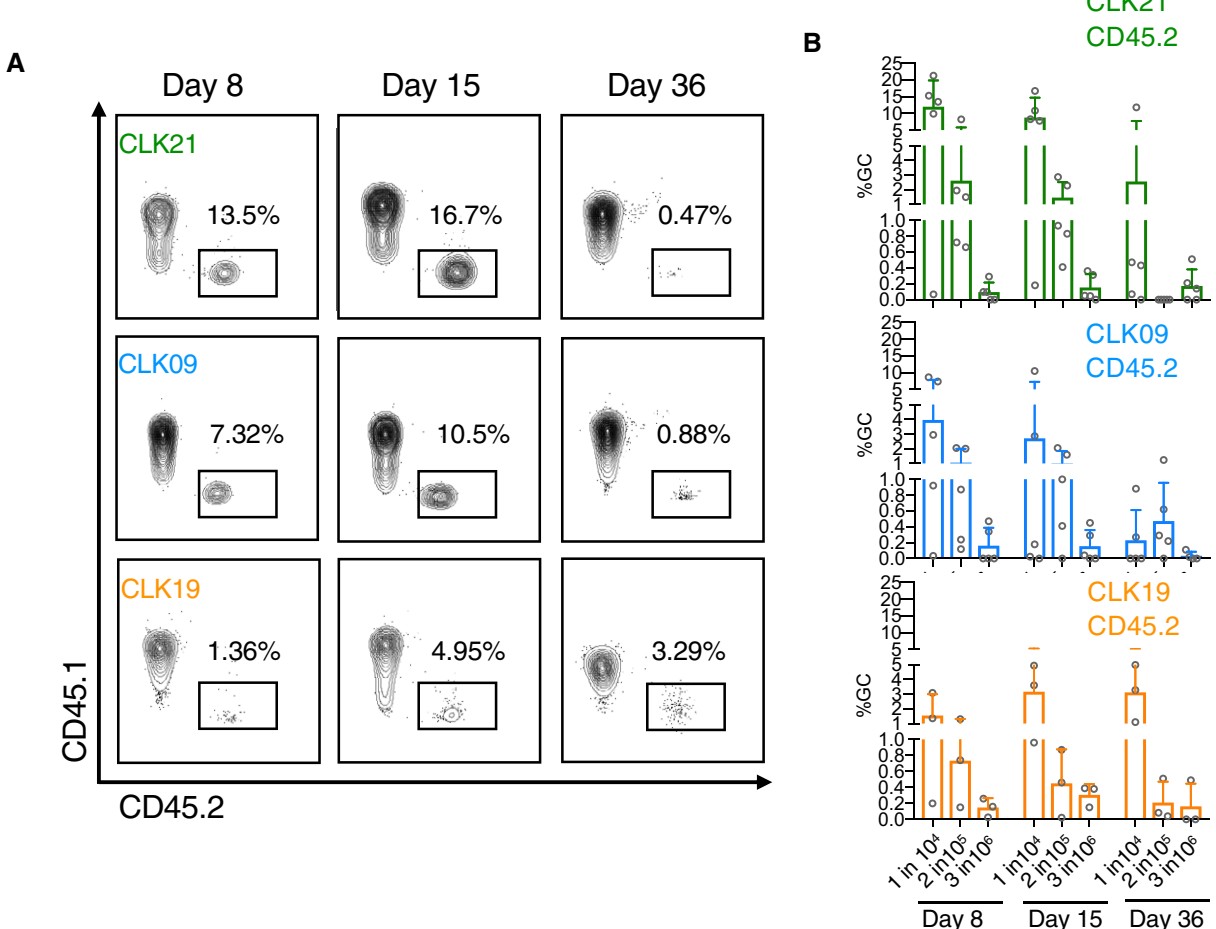

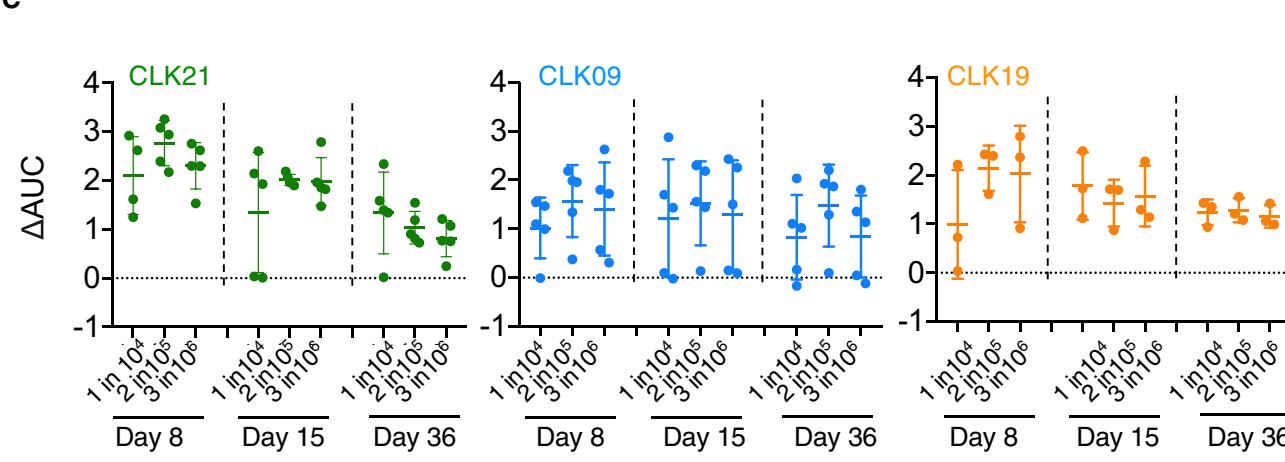

**Figure 6.**

CLK21 belongs to the PCIN63/Vκ1-5 sub-class of VRC01-class anti-bodies, whereas CLK19 and CLK09 are members of the N6/Vκ1-33 sub-class. PCIN63 bnAbs have a reduced level of neutralizing breadth compared to most other VRC01-class bnAbs, but the SHM frequency (9.6–16% and 10.0–13.7% nucleotide mutation for HC and LC) is also 2- to 3-fold lower than most VRC01-class bnAbs, which suggests that the PCIN63/ Vκ1-5 sub-class should be included as a target for vaccine design (Umotoy *et al*, 2019). The N6/Vκ1-33 sub-class includes one of the highest breadth bnAbs, N6, which can neutralize ~ 98% HIV-1 strains (Huang *et al*, 2016). We found that a

single immunization with eOD-GT8 60mer generated substantial sequence diversity and SHM, including substantial numbers of VRC01-class mutations in the VH1-2 gene and, in the two N6/Vκ1-33 models CLK09 and CLK19, accumulation of HCDR3 mutations (Fig 7). Importantly, we demonstrated that eOD-GT8 60mer can prime all three VRC01-class BCRs *in vivo* at low precursor frequencies approximating those measured in humans (Figs 5 and 6). Therefore, our results suggest that eOD-GT8 60mer could potentially induce such responses in humans.

In conclusion, this study not only reports a robust technology that may accelerate pre-clinical validation and iteration of immunogens but also hints at certain subclasses of VRC01-class B cells that might be stimulated by eOD-GT8 60mer in the human clinical trial. Indeed, our observation that VRC01-class naive precursor B cells carrying Vk1-33 and Vk1-5 light chains can be activated in the mouse by eOD-GT8 60mer, predicts that they might also be activated in humans. Future comparison of the results of this study with the results of the clinical trial will further assess the correlation if any between the two and will thus provide greater insight into the value of these types of KI mouse models. Thus, our work represents a significant breakthrough that not only accelerates the time for the pre-clinical validation and iteration of HIV immunogens but also is likely to be important in vaccine development against other infectious diseases.

# Materials and Methods

## Reagents and Tools table

| Reagent/Resource | Reference or source | Identifier or catalog number |
|---|---|---|
| **Experimental models** | | |
| Mouse: B6.SJL-Ptprcapepcb/BoyJ | The Jackson Laboratory | JAX: 002014 |
| Mouse: C57BL/6 | The Jackson Laboratory | JAX: 000664 |
| Mouse: PGT121 κ KI mouse model | This paper | N/A |
| Mouse: CLK21 BCR KI mouse model | This paper | N/A |
| Mouse: CLK09 BCR KI mouse model | This paper | N/A |
| **Antibodies** | | |
| PerCP/Cy5.5 anti-mouse CD45.1 Antibody | BioLegend | Cat#: 110728 |
| PE-Cy™7 Hamster Anti-Mouse CD95 | BD Biosciences | Cat#: 557653 |
| Brilliant Violet 785™ anti-mouse CD45.2 | BioLegend | Cat#: 109839 |
| anti-mouse CD38, BV510 | BD Biosciences | Cat#: 740129 |
| Pacific Blue™ anti-mouse/human CD45R/B220 Antibody | BioLegend | Cat#: 103227 |
| BV711 Rat Anti-Mouse Ig, λ1, λ2 & λ3 Light Chain | BD Biosciences | Cat#: 744527 |
| BUV395 Rat Anti-Mouse Ig, κ light chain | BD Biosciences | Cat#: 742839 |
| APC/Cy7 anti-mouse/human CD45R/B220 Antibody | BioLegend | Cat#: 103224 |
| BV421 Rat Anti-Mouse IgM | BD Biosciences | Cat#: 743323 |
| PE/Cy7 anti-mouse IgD Antibody | BioLegend | Cat#: 405720 |
| PE anti-mouse CD45.2 Antibody | BioLegend | Cat#: 109808 |
| Anti-mouse CD38 Alexa700 | Invitrogen | Cat#: 56-0381-82 |
| BV786 Rat Anti-Mouse IgD | BD Biosciences | Cat#: 563618 |
| BV510 Rat Anti-Mouse B220/CD45R Clone RA3-6B2 | BD Biosciences | Cat#: 563103 |
| BV421 Rat Anti-Mouse IgG1 | BD Biosciences | Cat#: 562580 |
| BUV395 Rat Anti-Mouse IgM | BD Biosciences | Cat#: 743329 |
| PE anti-mouse CD80 antibody | BioLegend | Cat#: 104708 |
| APC/Cyanine7 anti-mouse CD73 Antibody | BioLegend | Cat#: 127232 |
| GL7 Monoclonal Antibody (GL-7 (GL7)), eFluor 450 | Thermo Fisher | Cat#: 48-5902-82 |
| BUV395 Rat Anti-Mouse CD273 | BD Biosciences | Cat#: 565102 |
| CD4 Monoclonal Antibody (GK1.5), APC-eFluor 780 | Invitrogen | Cat#: 47-0042-80 |
| CD8a Monoclonal Antibody (53-6.7), APC-eFluor 780 | Invitrogen | Cat#: 47-0081-80 |
| F4/80 Monoclonal Antibody (BM8), APC-eFluor 780 | Invitrogen | Cat#: 47-4801-80 |
| Ly-6G Monoclonal Antibody (1A8-Ly6g), APC-eFluor 780 | Invitrogen | Cat#: 47-5931-80 |
| Alexa Fluor® 488 anti-mouse CD38 Antibody | BioLegend | Cat#: 102714 |

**Reagents and Tools table**  (continued)

| Reagent/Resource | Reference or source | Identifier or catalog number |
| --- | --- | --- |
| Alexa Fluor® 647 anti-mouse/human GL7 Antigen | BioLegend | Cat#: 144606 |
| Alexa Fluor® 594 anti-mouse/human CD45R/B220 Antibody | BioLegend | Cat#: 103254 |
| Alkaline Phosphatase AffiniPure Goat Anti-Mouse IgG, Fcγ Fragment Specific | Jackson ImmunoResearch | Cat#: 115-055-071 |
| Purified Rat Anti-Mouse CD16/CD32 (Mouse BD Fc Block™ | BD Biosciences | Cat#: 553142 |
| Alexa Fluor® 488 anti-mouse CD45.2 Antibody | BioLegend | Cat#: 109816 |
| Alexa Fluor® 594 anti-mouse CD3 Antibody | BioLegend | Cat#: 100240 |
| **Oligonucleotides and other sequence-based reagents** | | |
| TaqMan probes for genotyping | TransnetYX | Appendix Table S1 |
| Primers for RT–PCR | GENEWIZ | Appendix Table S6 |
| Index primers for NGS, | Integrated DNA Technologies | Appendix Table S7 |
| sgRNA | Synthego | Table 1 |
| **Chemicals, enzymes and other reagents** | | |
| LIVE/DEAD™ Fixable Blue Dead Cell Stain Kit, for UV excitation | Thermo Fisher Scientific | Cat#: L34962 |
| Alexa Fluor 488 Streptavidin | BioLegend | Cat#: 405235 |
| Alexa Fluor 647 Streptavidin | BioLegend | Cat#: 405237 |
| Alexa Fluor 594 Streptavidin | BioLegend | Cat#: 405240 |
| Pan B-Cell Isolation Kit II, mouse | Miltenyi Biotec | Cat#: 130-104-443 |
| SuperScript™ III Reverse Transcriptase | Thermo Fisher | Cat#: 18080085 |
| RNeasy® Protect Animal Blood Kit | Qiagen | Cat#: 73224 |
| Maxima First Strand cDNA Synthesis Kit | Thermo Fisher Scientific | Cat#: K1672 |
| PrimeSTAR® HS DNA Polymerase premix | Takarabio | Cat#: R040A |
| HIFI HOTSTART READYMIX 500RXN | Kapa Biosystems | Cat#: KK2602 |
| HotStarTaq DNA Polymerase | Qiagen | Cat#: 203205 |
| RNasin® Ribonuclease Inhibitors (Recombinant) | Promega | Cat#: N2515 |
| CountBright™ Absolute Counting Beads, for flow cytometry | Thermo Fisher Scientific | Cat#: C36950 |
| SIGMAFAST™ p-Nitrophenyl phosphate Tablets | Sigma | Cat#: N2770-50SET |
| NP40 | Millipore | Cat#: 492016-100ML |
| UltraComp eBeads™ Compensation Beads | Thermo Fisher Scientific | 01-2222-42 |
| Tissue-Tek® O.C.T. Compound, Sakura® Finetek | VWR | 25608-930 |
| eOD-GT8 60mer | Jardine et al (2015) | N/A |
| eOD-GT8 KO 60mer | Jardine et al (2015) | N/A |
| eOD-GT8 soluble protein | Sok et al (2016) | N/A |
| eOD-GT8 KO soluble protein | Sok et al (2016) | N/A |
| **Software** | | |
| Flowjo X | Treestar | https://www.flowjo.com/ |
| Prism 8 | GraphPad | https://www.graphpad.com/ |
| Microsoft Office | Microsoft | https://www.office.com/ |
| IMGT/V-quest | | http://www.imgt.org/IMGTindex/V-QUEST.php |
| Geneious Prime | Biomatters | https://www.geneious.com/ |
| **Other** | | |
| Master Cycler PCR machine | Eppendorf | |
| ELISA readers | BioTek | |
| Zeiss Elyra PS.1 confocal microscope | Zeiss | |
| 5 laser LSR Fortessa | BD | |

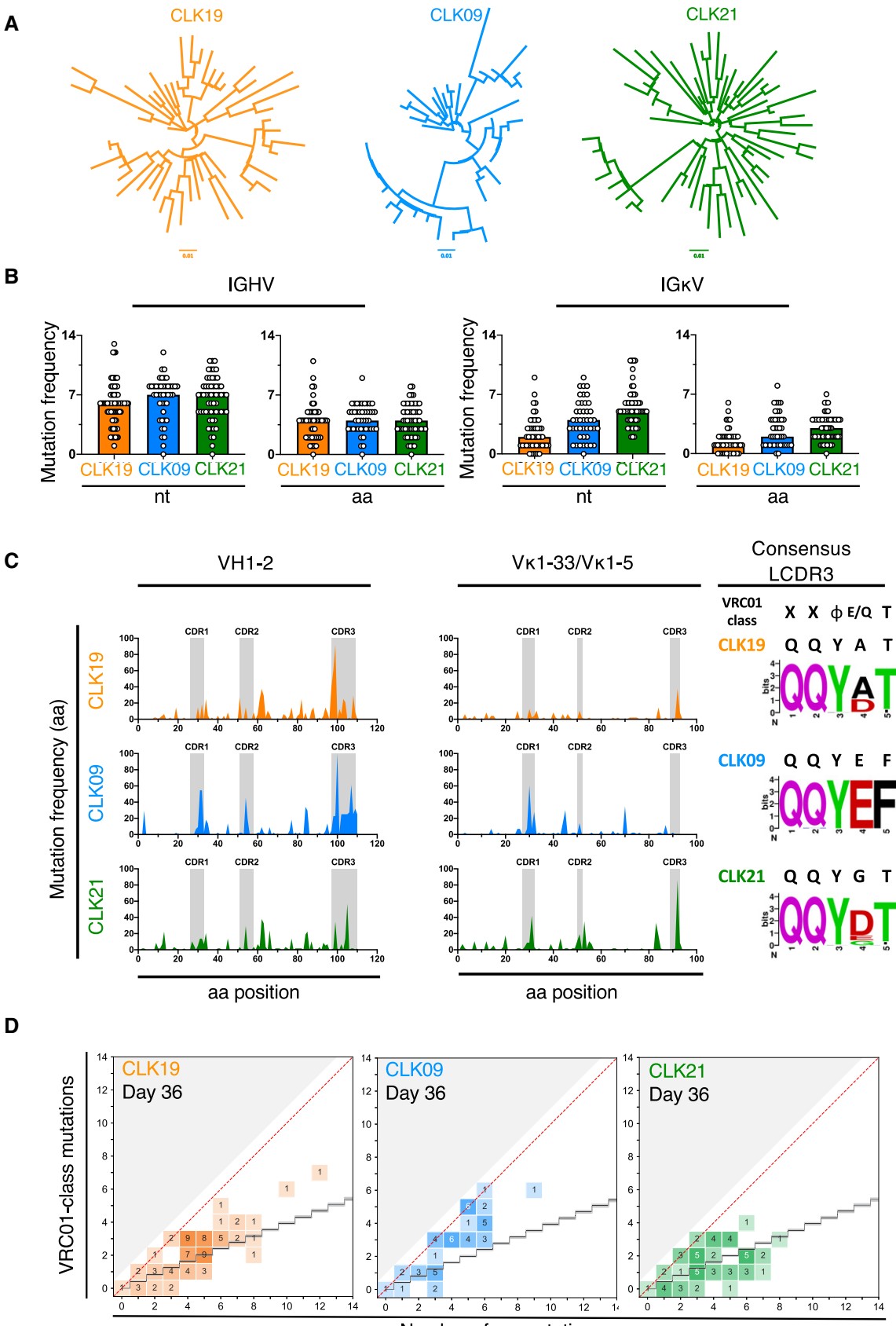

Figure 7.

**Figure 7. VRC01-class precursor B cells exhibit SHM and VRC01-like mutations 36 days after a single priming injection with eOD-GT8 60mer.**

Three CLK-mouse lines were generated and subsequently immunized with eOD-GT8 60mer in adoptive transfer/immunization system (precursor frequency: 1 in 10[4]). eOD-GT8-specific splenic IgG1[+] or IgM[−]IgD[−] B cells were single-cell sorted at Day 36 post-immunization for single-cell PCRs.

A  Phylogenetic trees of CLK21, CLK09 and CLK19. These trees were generated using paired aminoacidic sequences solely isolated at Day 36. Single paired amino acid sequences were joined and aligned using MUSCLE (Price *et al*, 2010). Clonal lineage trees were generated using FastTree and a Jones Taylor Thornton model for AA evolution (Jones *et al*, 1992). The length of the branches reflects sequence distance.

B  SHM are detectable in both IGHV and IGLV at Day 36 post-immunization with eOD-GT8 60mer. nt represents "nucleotide", aa represents "amino acid".

C  Hotspot analysis. The quality of mutations was assessed via hotspots analysis for both heavy (left) and light (right) antibody chains. Weblogos were generated via publicly available online tools (https://weblogo.berkeley.edu/logo.cgi).

D  Mature bnAbs-like aa VH1-2 mutations in CLK B cells at Day 36. The red diagonal line indicates a 100% efficiency of VRC01-class bnAb-type VH1-2 mutations. The black stair step indicates a calculated VH1-2 antigen-agnostic mutation distribution, which might include mutations that improve expression or stabilize Ab structure (Briney *et al*, 2016). Shaded area indicates that no mutation is positive in this region.

**Table 1. sgRNA sequence for PGT 121 κ KI mice.**

| No. | sgRNA sequence | PAM |
|-----|----------------|-----|
| K1 | CCACTGTGGTGGACGTTCGG | TGG |
| K3 | CTACCACTGTGGTGGACGTT | CGG |
| K4 | GTTCGGTGGAGGCACCAAGC | TGG |
| K6 | AGTGTGTGTACACGTTCGGA | GGG |
| K8 | GCCCTAGACAAACCTTTACT | CGG |
| K11 | GGGCTCATTATCAGTTGACG | TGG |
| K12 | CACCATCCAAGAGATTGGAT | CGG |
| K13 | TATTCTCCGATCCAATCTCT | TGG |
| K14 | CTGAGCGAAAAACTCGTCTT | AGG |
| K15 | CTGTGGCTCACGTTCGGTGC | TGG |
| K16 | TTTGGCCCATCTAGTTGGAC | TGG |
| K18 | TGTGATTCACGTTCGGCTCG | GGG |

## Methods and Protocols

### Protein production and purification

eOD monomers and 60mers were produced and purified as described previously (Jardine *et al*, 2013). Briefly, eOD 60mers for immunization were produced in the presence of 14 mM kifunensine in HEK293F or HEK293H cells (Invitrogen), and eOD monomers for ELISA were produced without kifunensine in HEK293F or HEK293H cells. The eOD-GT8 KO and eOD-GT8 KO 60mer constructs in this study were based on the improved eOD-GT8 KO2 design (Sok *et al*, 2016). For simplicity, eOD-GT8 KO2 is referred to as eOD-GT8 KO in the text and figures.

Avi-tagged eOD-GT8 and eOD-GT8 KO monomers were purified and biotinylated as previously described (Sok *et al*, 2016). These constructs were individually premixed with fluorescently labeled streptavidin to form two positive tetramer probes (Alexa Fluor 488-eOD-GT8 and Alexa Fluor 647-eOD-GT8) and one negative tetramer probe (Alexa Fluor 594-eOD-GT8 KO). These three independent fluorochromes were chosen to detect eOD-GT8 epitope-specific B cells.

### Mice and immunization

C57BL/6 (CD45.2[+/+]) mice purchased from Jackson Lab were used for KI mice generation as previously described (Lin *et al*, 2018).

Initial mice breeding of F0 generation was performed at the animal facility of the Gene Modification Facility (Harvard University). Breeding for colony expansion and experimental procedures was performed in the Ragon Institute. Ear or tail snips were used for genotyping. All of the mice were genotyped by TaqMan assay for a fee for service agreement (TransnetYX). TaqMan probes for the genotyping assay were developed by TransnetYX as shown in Appendix Table S1. 7–12-week-old of male B6.SJL-Ptprcapepcb/BoyJ mice (CD45.1[+/+]) were purchased from Jackson Lab for experiment. Isolated CD45.2 B cells by Pan B-Cell Isolation Kit II (Miltenyi Biotec) from CLK21,CLK09 and CLK19 BCR KI mice were adoptively transferred to the CD45.1 mice as previously reported (Abbott *et al*, 2018). After 12 h, each mouse was immunized intraperitoneally (i.p.) with 15 μg of eOD-GT8 60mer or eOD-GT8 KO 60mer in 100 μl of PBS plus 2% Alhydrogel (Invitrogen) in 100 μl of PBS. On Day 8, 15, and 36, the splenocytes were isolated and detected by FACS. All experiments were approved by the Institutional Animal Care and Use Committee (IACUC) of Harvard University and the Massachusetts General Hospital and conducted in accordance with the regulations of the American Association for the Accreditation of Laboratory Animal Care (AAALAC).

### sgRNA selection and validation

sgRNAs PGT121 κ KI mice were identified using the CRISPR DESIGN database (https://zlab.bio/guide-design-resources) (Table 1) and purchased from Synthego. PCR amplification of genomic DNA fragments from C57BL/6 control mice was performed by using gene-specific forward primer(5′-GACAGTTGTGAGC-TACTTAT) and reverse primer (5′-CACGGATGAGTCTCCTTCTC) under the following conditions: 98°C for 2 min; 40 × (98°C for 20 s, 55°C for 30 s and 72°C for 1 min); 72°C for 5 min; hold at 4°C. sgRNA validation was performed by using in vitro digestion with Cas9 nuclease, *S. pyogenes* assay (NEB) following the manufacturer's instructions and our previous publication (Lin *et al*, 2018). sgRNAs for CLK21, CLK09 and CLK19 KI mice were same to PGT121 κ and BG18[gH] that was reported in our previous paper (Lin *et al*, 2018).

### Generation of targeting vectors

The 4E10 light chain vector backbone (Ota *et al*, 2013) was modified by replacing the neomycin cassette with rearranged VJ sequences for human bnAb germline precursors PGT121 κ downstream of the promoter region and by elongation of the 5′ and 3′ homology regions utilizing the Gibson assembly method (NEB). The targeting vector DNA was purified with EndoFree Plasmid

Maxi Kit (Qiagen) and confirmed by Sanger sequencing (Eton Bioscience Inc). The vectors for CLK21, CLK09 and CLK19 were same to PGT121 κ and BG18[gH] that was reported in our previous paper (Lin *et al*, 2018).

### Generation of KI mice

Light chain single injection: Two sgRNAs, donor DNA and Cas9 protein (PNA BIO INC) were mixed together for the generation KI mice by CRISPR/Cas9 method (Lin *et al*, 2018). Heavy and Light chains double injection: Briefly, we prepared the injection mix of Cas9 protein (100 ng/μl, IDT), sgRNA (50 ng/μl, Synthego) and circular DNA donor (10 ng/μl) in injection buffer as previous papers described (Yang *et al*, 2013; Yang *et al*, 2014) and performed microinjecting 200 zygotes. After culturing the injected zygotes, we transferred around 30 of the zygotes into the oviducts of 5–7 (average 6) of pseudo-pregnant foster mothers.

### Flow cytometry

To evaluate the GC response induced by eOD-GT8 60mer in adoptively transferred mice, spleens were isolated and crushed with 5 ml of Syringe in FACS buffer (2% FBS in PBS), red blood cells were removed by incubating with 2 ml of ACK lysis buffer (VWR/Lonza) for 3 min. Splenocytes were resuspended in FACS buffer followed by stained with 500-fold diluted Live/Dead Blue(Thermo Fisher Scientific) for 20 min at 4°C. 100-fold diluted anti-mouse CD16/CD32 Fc block was added and incubated for 20 min at 4°C. Alexa Fluor 488 and Alexa Fluor 647 conjugated eOD-GT8 tetramer probe and Alexa Fluor 594 conjugated eOD-GT8KO probe were added and incubated for 20 min at 4°C. 100-fold diluted antibodies testing for B-cell surface markers were incubated for 20 min at 4°C. After washed with FACS buffer once, the cells were filtered and resuspended in FACS tubes and detected by 5 Laser LSR Fortessa. To sort the antigen-specific CD45.2 B cells, after stained with probes as mentioned above, the cells were incubated with 100-fold diluted B-cell surface markers. Single-cell plates were sorted by SORP Aria II machine and gated as SSL[+], B220[+], eOD-GT8[+](A488 and A647[+]), IgM[−], IgD[−], IgG1[+], eOD-GT8KO-(A594[−]).

### Immunofluorescence

Spleens were frozen in Tissue-Tek OCT Compound (VWR) and cryosectioned. Sections were fixed with 4% PFA for 10 min, washed three times in PBS, blocked 45 min with 10% goat serum in PBS, and stained 1 h with 100-fold diluted antibodies (Pacific blue anti-mouse B220, Alexa Fluor 488 anti-mouse CD45.2, Alexa Fluor 594 anti-mouse CD3 and Alexa Fluor 647 anti-mouse GL7). After washing three times with PBS, stained sections were mounted with ProLong Diamond Antifade Mountant (Thermo Fisher). Images were acquired on a Zeiss Elyra PS.1 confocal microscope and analyzed in ImageJ 2.0.0.

### ELISA

To assess the IgG titers of immunized mice, 96-well ELISA plates were pre-coated with eODGT8 or eOD-GT8KO at a concentration of 50 ng per well. Pre-coated plates were incubated at 4°C overnight. After incubation with blocking buffer (3% BSA in PBS + 0.01% Tween 20) for 2 h at RT, 3-fold diluted sera from eOD-GT8 60mer or eOD-GT8 KO 60mer immunized mice were incubated with pre-coated protein for 2 h at RT. This was followed by incubation with 3000-fold diluted Alkaline Phosphatase AffiniPure Goat Anti-Mouse

IgG for 1 h at RT. Finally, 50 μl per well of 1 mg/ml p-Nitrophenyl phosphate dissolved in ddH$_2$O was added and incubated for 20 min at RT for chromogenic reaction. OD405 was read using an ELISA reader (BioTek).

### Single-cell PCR

In order to assess antibody evolution and mutation frequency, single-cell PCR was performed using sorted B cells in 96-well plates. Briefly, cDNA was prepared by RT–PCR with SuperScriptTM III Reverse Transcriptase kit (Thermo Fisher), the IgG and Ig Kappa were amplified as previously reported (von Boehmer *et al*, 2016).Finally, Sanger sequencing was performed by GENEWIZ company.

### Phylogenetic analysis

Single paired amino acid sequences were joined and aligned using MUSCLE. Clonal lineage trees were generated using FastTree (Price *et al*, 2010) and a Jones Taylor Thornton model for AA evolution (Jones *et al*, 1992).

### Statistical analysis

Significant differences were calculated with Student's *t*-test and shown as: $P > 0.05$, no statistical significance (ns), *$P < 0.05$, **$P < 0.01$, ***$P < 0.001$, ****$P < 0.0001$. All *P* value analyses were calculated by GraphPad Prism V8.0.

## Data availability

This study includes no data deposited in external repositories.

**Expanded View** for this article is available online.

## Acknowledgements

We would like to thank the members of the FACS and microscopy facilities at Ragon Institute for outstanding expertise. We also thank T. Schiffner for critical reading of the manuscript. This work was supported by the National Institute of Allergy and Infectious Diseases (NIAID) UM1 AI100663 (Scripps Center for HIV/AIDS Vaccine Immunology and Immunogen Discovery) and UM1 AI144462 (Scripps Consortium for HIV/AIDS Vaccine Development) (to W.R.S., F.D.B., S.C.); by the Bill & Melinda Gates Foundation OPP1214085 (to F.D.B.); by the Ragon Institute of MGH, MIT, and Harvard (to F.D.B., W.R.S.); by the International AIDS Vaccine Initiative (IAVI) Neutralizing Antibody Consortium (NAC) and Center (to W.R.S.); and through the Collaboration for AIDS Vaccine Discovery funding for the IAVI NAC Center (to W.R.S.).

## Author contributions

FDB conceived this study. FDB and WRS provided the project outline and were instrumental in the design of experiments. XW and RR helped design and perform experiments, analyzed the data, and together with UN, KHK, SK, EM, and FDB composed the manuscript. UN together with DC and LW designed and implemented the CRISPR approach. SK together with EM, Y-CL performed the experiments related to PGT121 κ mouse model. SK together with EM, JW, AL performed the sequencing analysis. SG performed immunohistochemistry staining and microscopy. KHK and LX coordinated mouse sample collection and work. BG, NP, YA, RT prepared the immunogen-related reagents. SC provided the human sequence. All of the authors provided critical feedback on the manuscript prior to publication and have agreed to the final content.

## Conflict of interest

The authors declare that they have no conflict of interest.

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
