## [Review Process File · The EMBO Journal]

Multiplexed CRISPR/CAS9-Mediated Engineering of Pre-clinical Mouse Models Bearing Native Human B Cell Receptors

Xuesong Wang, Rashmi Ray, Sven Kratochvil, Eleonora Melzi, Ying-Cing Lin, Sophie Giguere, Liling Xu, John Warner, Diane Cheon, Alessia Liguori, Bettina Groschel, Nicole Phelps, Yumiko Adachi, Ryan Tingle, Lin Wu, Shane Crotty, Kathrin Kirsch, Usha Nair, William Schief and Facundo Batista
DOI: [10.15252/emboj.2020105926](https://doi.org/10.15252/emboj.2020105926)

Corresponding author(s): Facundo Batista (fbatista1@mgh.harvard.edu), William Schief (schief@scripps.edu)

Review Timeline:

Submission Date:	15th Jun 20
Editorial Decision:	22nd Jul 20
Revision Received:	13th Oct 20
Accepted:	23rd Oct 20

Editor: Karin Dumstrei

Transaction Report:

Dear Facundo,

Thanks for submitting your manuscript to The EMBO Journal. Your study has now been seen by two referees who also had access to the referee comments from a previous journal as well as your response.

As you can see from the comments below, the referees feel a bit mixed if the added value of targeting simultaneous IgH and IgL loci using CRISPR/Cas9 methodology is enough of a technological advance in order to consider publication here. While Referee #2 is supportive of the work and find it an important step forward, referee #1 is a bit more hesitant if we gain enough new insight.

I have discussed the referee comments further with my colleagues and while we do see the point raised by referee #1 we also find the described approach useful and of value to the field. I would therefore like to invite you to submit a revised version - only text changes and clarifications are needed.

Regarding the previous referee comments - do you have data on hand to address the comment raised by referee #1 to including SHM analyses for the lowest precursor frequency... (2nd to last point). If you have this data then it would be good to include it.

That should be all. Let me know if we need to discuss anything further.

best Karin

Karin Dumstrei, PhD
Senior Editor
The EMBO Journal

When assembling figures, please refer to our figure preparation guideline in order to ensure proper formatting and readability in print as well as on screen:
<http://bit.ly/EMBOPressFigurePreparationGuideline>

IMPORTANT: When you send the revision we will require
- a point-by-point response to the referees' comments, with a detailed description of the changes

made (as a word file).

- a word file of the manuscript text.

- individual production quality figure files (one file per figure)

- a complete author checklist, which you can download from our author guidelines (<https://www.embopress.org/page/journal/14602075/authorguide>).

- Expanded View files (replacing Supplementary Information)

Further information is available in our Guide For Authors:

The revision must be submitted online within 90 days; please click on the link below to submit the revision online before 20th Oct 2020.

Referee #1:

The manuscript by Bautista and colleagues describes a methodology to generate mice with knock-in V-region sequences in both the heavy and light chain loci. The technology uses CRISPR/Cas9 and leads to efficient modification of the endogenous IgH and IgL loci. Such B cells respond well to antigen, undergo CSR and somatically mutate their V regions. The paper is clear, well written, and the results are convincing.

My only major concern is the novelty and the main conceptual advance being reported here? Knocking in sequences with CRISPR/Cas9 is routinely used by many, including commercial sources, and was previously reported by the authors to knockin V-regions into the IgH locus. What is being reported here is that they can accomplish this at both the IgH and IgL loci simultaneously. In addition, it has already been reported by many groups that mice with pre-rearranged V-regions that have been knocked into the IgH locus undergo CSR and SHM.

While these mice will represent a good tool to perhaps develop more potent bNAbs, or study the biological response to HIV in mice, the technological advance, at least to me, is not very striking.

Referee #2:

The study of Wang et is technically very sound, theoretically and practically interesting and will be

of significant interest to those studying B cell biology, particularly against complex immunogens that are ultimately designed for human use. I note also that the authors have made many revisions in light of 3 thorough reviews. While I am reluctant to add to their existing efforts, there are three things I would suggest they address for clarity.

1. Lines 359-360. This was quite confusing as the wording here suggests the transferred frequency was 1 in 10^4 CD45.2 cells, with the frequency of eOD-GT8 binding cells within this group being variably 14-63% depending on the genotype of the relevant B cells (line 317-318). Later, however, this experimental procedure is written in such a way that it infers the frequency of eOD-specific B cells is 1 in 10^4 etc. This should be clarified to be either strictly referring to CD45.2 cells or actually calculate the frequencies of the antigen specific B cells for the different naïve populations that were transferred. These values could influence the kinetics of the resultant responses, whereas now it could appear to the casual reader that the same number of precursors were used for each construct.

2. Line 393, (Figure S7A is missing a closing parenthesis.

3. Line 432 "silent to silent-mutations". One of those should be "non-silent".

4. "accruement" is a most unusual word. Would "accrual" have wider recognition?

5. Line 452 missing a comma after (Jardine et al., 2016b)

6. Line 410-411, it is potentially misleading to say that IgG1 cells appeared in the GC at d36, when no assessment was done between days 15 and 36 and only day 36 data are shown. Please be more precise in referring to what was done - IgG1 cells were only assessed on day 36? IgG1 cells were absent on day 15? It is a small but important point.

7. How many samples were analyzed for the data in S2 and S3? Statistics would be best, but in their absence the number of samples of which this is representative should be given.

Point by point response to the Reviewers

Reviewer#1(R1):

The manuscript by Batista and colleagues describes a methodology to generate mice with knock-in V-region sequences in both the heavy and light chain loci. The technology uses CRISPR/Cas9 and leads to efficient modification of the endogenous IgH and IgL loci. Such B cells respond well to antigen, undergo CSR and somatically mutate their V regions. The paper is clear, well written, and the results are convincing.

My only major concern is the novelty and the main conceptual advance being reported here? Knocking in sequences with CRISPR/Cas9 is routinely used by many, including commercial sources, and was previously reported by the authors to knockin V-regions into the IgH locus. What is being reported here is that they can accomplish this at both the IgH and IgL loci simultaneously. In addition, it has already been reported by many groups that mice with pre-rearranged V-regions that have been knocked into the IgH locus undergo CSR and SHM.

While these mice will represent a good tool to perhaps develop more potent bnAbs, or study the biological response to HIV in mice, the technological advance, at least to me, is not very striking.

Response: We thank R1 for his/her positive comments on our manuscript highlighting that “*The paper is clear, well written, and the results are convincing*”. R1 expressed some concerns regarding the novelty to which we respectfully disagree for the following reasons: As we have mentioned in our Discussion section, line 501-505, to the best of our knowledge, ours is the first *in vivo* report of multiple HDR-mediated targeted insertions – this is unprecedented

1. As far as B cell repertoire knock-in (KI) mice are concerned, although CRISPR/Cas9 has been used to generate mice bearing pre-rearranged V-regions into the Ig H locus, ours is the first report of CRISPR/Cas9-mediated KI of LC immunoglobulin variable regions into the κ locus of fertilized zygotes. Previously, this has proved to be very difficult.
2. Our CRISPR/Cas9-mediated LC KI models are novel because in the past insertions at the κ locus have either relied on ES cells (Pelanda et al., Immunity, 1997) or non-physiological approaches such as the targeting of bicistronic Ig H and L chains to the Ig H locus (Jacobsen et al., Journal of Experimental Medicine, 2018).
3. In regard to the relevance of the VRC01-class KI mice to the HIV field, it is known that for this class of bnAbs it is of critical importance to knock in both HC and LC sequences at their native loci to study response to germline targeting immunogens. This is because while some HIV-1 bnAbs only require HCs to neutralize the virus, VRC01 bnAbs are dependent on both HC and LCs. For example, in the VRC01 subclass bnAbs, a 5-aa LC is indispensable for bnAb function. Thus it cannot be more strongly emphasized that mouse models with concomitant HC and LC VRC01-subclass KI sequence expression at their native loci are necessary for the studies of CSR, SHM and antibody evolution in response to immunization.
4. Finally, the mouse models reported here are based on the KI of bona fide native human BCR sequences, which are important in order to assess how these B cells will respond to germline-targeting immunogens. In contrast, in the past testing germline-targeting immunogens has often relied on KI mouse models expressing inferred germline (iGL) reverted precursors of known bnAbs.

Taken together, we believe that the scientific contributions of this work are significant not only in terms of multiplexed *in vivo* HDR KI mouse generation, but also in providing valuable preclinical BCR KI models for immunogen evaluation.

Reviewer#2 (R2):

The study of Wang et al is technically very sound, theoretically and practically interesting and will be of significant interest to those studying B cell biology, particularly against complex immunogens that are ultimately designed for human use. I note also that the authors have made many revisions in light of 3 thorough reviews. While I am reluctant to add to their existing efforts, there are three things I would suggest they address for clarity.

Response: We thank R2 for his/her positive comments pointing out that our manuscript is “*is technically very sound, theoretically and practically interesting and will be of significant interest to those studying B cell biology*”. We also thank R2 for her/his constructive suggestions to improve clarity and readability of our manuscript which we have fully address it below:

Major comments:

R2 states: *Lines 359-360. This was quite confusing as the wording here suggests the transferred frequency was 1 in 10^4 CD45.2 cells, with the frequency of eOD-GT8 binding cells within this group being variably 14-63% depending on the genotype of the relevant B cells (line 317-318). Later, however, this experimental procedure is written in such a way that it infers the frequency of eOD-specific B cells is 1 in 10^4 etc. This should be clarified to be either strictly referring to CD45.2 cells or actually calculate the frequencies of the antigen specific B cells for the different naïve populations that were transferred. These values could influence the kinetics of the resultant responses, whereas now it could appear to the casual reader that the same number of precursors were used for each construct.*

Response: We apologize for the confusion regarding the cell frequency used and thank R2 for pointing this out. In our manuscript, all the transferred frequencies were calculated as described in experimental procedure as (Number of CD45.2 binders)/(Number of B cells). We have now corrected it in our revised manuscript. Please see line 362.

R2 states: *Line 393, (Figure S7A is missing a closing parenthesis.*

Response: We have added the missing parenthesis in line 396.

R2 states: *Line 432 "silent to silent-mutations". One of those should be "non-silent".*

Response: We have corrected it in our revised manuscript. Please see line 439.

R2 states: *"accruement" is a most unusual word. Would "accrual" have wider recognition?*

Response: As suggested, we have replaced accruement with accrual, line 448.

R2 states: *Line 452 missing a comma after (Jardine et al., 2016b)*

Response: We have added the missing comma, line 461.

R2 states: *Line 410-411, it is potentially misleading to say that IgG1 cells appeared in the GC at d36, when no assessment was done between days 15 and 36 and only day 36 data are shown. Please be more precise in referring to what was done - IgG1 cells were only assessed on day 36? IgG1 cells were absent on day 15? It is a small but important point.*

Response: We thank R2 for pointing this out. We assessed IgG1 cells only on day 36 and not on day 15. We have corrected this information to draw a precise conclusion in this revised version. Please see line 414-416.

***R2 states:** How many samples were analyzed for the data in S2 and S3? Statistics would be best, but in their absence the number of samples of which this is representative should be given.*

Response: A representative F1 breeder from each line was tested for T and B lymphocyte development and the positive progenies from the breeder were screened for persistent breeding. We have added this information in figure legends of S2 and S3 in Appendix.

Dear Facundo,

Thanks for submitting your revised manuscript.

I have now had a chance to take a careful look at the revised version and I appreciate the introduced changes and the inclusion of the SHM analyses for the lowest precursor frequency (Appendix Figure S10). I am therefore very pleased to accept the manuscript for publication here.

Congratulations on a very valuable resource!

With best wishes

Karin

Karin Dumstrei, PhD
Senior Editor
The EMBO Journal

Please note that it is EMBO Journal policy for the transcript of the editorial process (containing referee reports and your response letter) to be published as an online supplement to each paper. If you do NOT want this, you will need to inform the Editorial Office via email immediately. More information is available here: https://emboj.embopress.org/about#Transparent_Process

Your manuscript will be processed for publication in the journal by EMBO Press. Manuscripts in the PDF and electronic editions of The EMBO Journal will be copy edited, and you will be provided with page proofs prior to publication. Please note that supplementary information is not included in the proofs.

Should you be planning a Press Release on your article, please get in contact with embojournal@wiley.com as early as possible, in order to coordinate publication and release dates.

If you have any questions, please do not hesitate to call or email the Editorial Office. Thank you for your contribution to The EMBO Journal.

Corresponding Author Name: Facundo D. Batista

Manuscript Number: EMBOJ-2020-105926